# sciLaMA: A Single-Cell Representation Learning Framework to Leverage Prior Knowledge from Large Language Models

Hongru Hu [1][2]   Shuwen Zhang [3]   Yongin Choi [1][2]   Venkat S. Malladi [4]   Gerald Quon [1][2]

## Abstract

Single-cell RNA sequencing (scRNA-seq) enables high-resolution exploration of cellular diversity and gene regulation, yet analyzing such data remains challenging due to technical and methodological limitations. Existing task-specific deep generative models like Variational Auto-Encoder (VAE) and its variants struggle to incorporate external biological knowledge, while transformer-based foundational large Language Models (LLMs or large LaMs) face limitations in computational cost and applicability to tabular gene expression data. Here, we introduce sciLaMA (single-cell interpretable Language Model Adapter), a novel representation learning framework that bridges these gaps by integrating static gene embeddings from multimodal LLMs with scRNA-seq tabular data through a paired-VAE architecture. Our approach generates context-aware representations for both cells and genes and outperforms state-of-the-art methods in key single-cell downstream tasks, including batch effect correction, cell clustering, and cell-state-specific gene marker and module identification, while maintaining computational efficiency. sciLaMA offers a computationally efficient, unified framework for comprehensive single-cell data analysis and biologically interpretable gene module discovery. Source code is available at `https://github.com/microsoft/sciLaMA`

*Equal contribution [1]Department of Molecular and Cellular Biology, University of California, Davis, CA USA [2]Genome Center, University of California, Davis, CA USA [3]Department of Quantitative Health Sciences, Mayo Clinic, Rochester, MN USA [4]Health Futures, Microsoft Research, Redmond, WA USA. Correspondence to: Hongru Hu <hrhu@ucdavis.edu>, Venkat S. Malladi <vmalladi@microsoft.com>, Gerald Quon <gquon@ucdavis.edu>.

*Proceedings of the 42nd International Conference on Machine Learning*, Vancouver, Canada. PMLR 267, 2025. Copyright 2025 by the author(s).

## 1. Introduction

Single-cell RNA sequencing (scRNA-seq) has revolutionized studies of cellular heterogeneity and transcriptome dynamics by providing gene expression profiles at single-cell resolution. Deep generative models, particularly Variational Autoencoders (VAE) (Kingma & Welling, 2014) and its variants, have become widely used for analyzing scRNA-seq data, which enable dimensionality reduction and representation learning by projecting cells from high-dimensional gene spaces to lower-dimensional embedding spaces (Grønbech et al., 2020; Lopez et al., 2018). These cell embeddings facilitate downstream tasks such as cell clustering, trajectory inference, and differential expression analysis (Chen et al., 2021; Kana et al., 2023; Yan et al., 2023). VAE's nonlinear representation capabilities allow them to effectively model complex cellular landscapes, making them well-suited for tabular data like gene expression matrices. However, scRNA-seq analysis remains challenging due to technical noise, sparse measurements, and batch effects, which often obscure true biological signals (Lähnemann et al., 2020). Incorporating external prior knowledge of genes, such as their functional annotations or molecular sequence data, has the potential to mitigate these challenges. However, the representation of input gene expression data as fixed-length vectors in traditional VAEs such as scVI-tools (Lopez et al., 2018) is not directly compatible with the different representations of prior gene knowledge, such as variable-length molecular sequences or text descriptions. This prevents prior gene knowledge from being directly incorporated into traditional VAE architectures.

Large Language Models (LLMs), on the other hand, have emerged as powerful tools for learning gene representations from various sources, including literature-based textual descriptions (Chen & Zou, 2024; Liu et al., 2023), molecular sequences (Elnaggar et al., 2022; Lin et al., 2023), and large-scale expression atlases (Cui et al., 2024; Theodoris et al., 2023). These models encode sequential data through tokenization and transformer architectures to create static gene embeddings that capture rich biological information. However, LLMs also face challenges: they are computationally expensive to train and inherently less suited for processing tabular data such as gene expression matrices, where VAEs

demonstrate superior performance (Kedzierska et al., 2023).

To bridge the complementary strengths of VAEs and LLMs, we propose **sciLaMA** (**s**ingle **c**ell **i**nterpretable **La**nguage **M**odel **A**dapter), a novel representation learning framework that extends the siVAE (Choi et al., 2023) architecture to integrate precomputed static gene embeddings from pretrained multimodal LLMs with scRNA-seq tabular data. By combining the representation power of VAEs with the adaptable and knowledge-rich embeddings of LLMs, our approach projects static gene information into context-aware representations by aligning each dimension of gene and cell latent space within the unified paired-VAE framework (Section 3). This approach presents a unified framework that improves over state of the art methods in single-cell analysis in three tasks: (1) cell representation learning and batch effect correction, (2) gene expression imputation, and (3) discovery of biologically meaningful gene modules and cell-state-specific regulatory dynamics, all while maintaining computational efficiency.

**Contributions**: (1) We introduce a novel framework that incorporates external gene knowledge from pretrained LLMs with scRNA-seq data, facilitating context-aware cell and gene representation learning. (2) We demonstrate that our approach reduces computational requirements while improving performance compared to existing state-of-the-art methods across various single-cell tasks.

## 2. Related work

**Deep generative approaches for single cell analysis.** Deep generative models, particularly those based on variational autoencoders (VAEs), have advanced single-cell RNA sequencing (scRNA-seq) analysis. Methods such as scVI-tools learn low-dimensional cell embeddings for cell-centric tasks such as visualization, clustering, and batch correction. Researchers have also further utilized feature attribution techniques to identify important genes in specific cell populations and infer gene modules (Janizek et al., 2023) by leveraging the learned cell embeddings. However, these approaches primarily focus on cell representations without inferring gene representations, and pipelines leveraging other tools are needed for gene-centric analyses such as marker identification and gene network discovery. To address this limitation, siVAE (Choi et al., 2023) introduced a unified framework for learning both cell and gene representations, enabling direct gene-centric analyses using the gene representations and therefore eliminating the need for explicit gene module inference via external tools. However, siVAE gene representation learning involves training an encoder whose number of input nodes scales with the number of cells, thus limiting its applications to large datasets. Moreover, scVI, siVAE, and most other VAE methods do not integrate external knowledge into scRNA-seq analysis due

to the representational challenges discussed above. Methods such as GLUE (Cao & Gao, 2022) incorporate external information about regulatory interactions among features in the form of feature variables, however, such approaches struggle to utilize information such as molecular sequences or natural language descriptions of genes.

**Single-cell foundation language models.** Transformer-based large language models (LLMs) have recently been applied for single-cell data analysis. Unlike VAE-based methods, which treat scRNA-seq data as a cell-by-gene matrix, models such as scGPT (Cui et al., 2024) represent expression profiles as sequences of tokens, drawing similarities to natural language and demonstrating a novel way to represent single-cell data. However, despite their promise, single-cell LLMs exhibit certain limitations. Their performance in zero-shot scenarios is often unreliable, and fine-tuning them requires extensive computational resources and technical expertise compared to task-specific small models (Kedzierska et al., 2023). These drawbacks emphasize the need for approaches that are computationally efficient and capable of bridging foundational knowledge with real-world single-cell tasks.

**Applications of static gene embeddings in single-cell analysis**. Gene embeddings derived from non-single-cell biological data modalities can complement information derived from single-cell data analysis. For instance, precomputed gene embeddings from protein language models (PLMs), such as ESM and ProtTrans (Elnaggar et al., 2022; Lin et al., 2023), capture gene molecular properties and have been applied in frameworks like SATURN (Rosen et al., 2024) to identify conserved master regulatory genes across species. Similarly, models such as GenePT (Chen & Zou, 2024) and scELMo (Liu et al., 2023) use embeddings derived from textual descriptions of gene functions and biological pathways via natural language models such as OpenAI text-embedding model (OpenAI, 2022). These applications demonstrate the feasibility of incorporating external static gene embeddings from various modalities into single-cell analysis frameworks. By integrating such embeddings, researchers are able to improve the robustness of single-cell analysis, facilitate gene module characterization, and uncover regulatory dynamics.

## 3. Methods

Conceptually, sciLaMA is an adapter framework that integrates pretrained gene embeddings from LLMs of different modalities, and tailors them for downstream single-cell analyses. Instead of learning gene representations de novo, sciLaMA adapts and contextualizes these precomputed static gene embeddings by incorporating context specific cell-level data (cell-by-gene expression matrix). In this section, we detail the technical components of the sciLaMA framework

and its application to single-cell analysis.

## 3.1. Input data processing and notation

The sciLaMA framework requires two inputs: (1) A set of gene expression inputs $\{c_i\}_{i=1}^N$, representing scRNA-seq data for $N$ cells (scaled log-normalized expression) drawn from a specific cell population. Each of the $N$ cell vectors $c_i$ has $M$ measurements corresponding to individual genes. (2) Static gene embeddings $\{g_j\}_{j=1}^M$, derived from a single pre-trained language model (LaM). These embeddings provide $D$-dimensional representations of $M$ genes, capturing their properties derived from external prior knowledge, where the number $D$ depends on the embedding dimensionality of the specific LLM.

## 3.2. sciLaMA architecture

sciLaMA is based on a paired encoder-decoder design, inspired by siVAE (Choi et al., 2023), a interpretable deep generative model that jointly learns sample (cell) and feature (gene) embeddings using a paired VAE design. siVAE only uses scRNA-seq data to learn both sets of embeddings, whereas sciLaMA uses external data to inform gene embeddings. sciLaMA consists of two encoder-decoders: one for cells and one for genes (Figure 1a).

### 3.2.1. CELL ENCODER AND DECODER

The cell encoder $f_{\phi^{\text{cell}}}^{\text{cell}}(\cdot)$ projects each cell $i$'s expression profile $c_i$, represented as an $M$-dimensional gene expression vector, to parameters of a $K$-dimensional variational posterior distribution with mean $\boldsymbol{\mu}_i^{\text{cell}} \in \mathbb{R}^k$ and variance $(\boldsymbol{\sigma}_i^{\text{cell}})^2 \in \mathbb{R}^k$. A latent embedding $z_i^{\text{cell}}$ is sampled via the reparameterization trick:

$$
\begin{aligned}
\left(\boldsymbol{\mu}_i^{\text{cell}}, \boldsymbol{\sigma}_i^{\text{cell}}\right) &\leftarrow f_{\phi^{\text{cell}}}^{\text{cell}}(c_i) \\
z_i^{\text{cell}} &= \boldsymbol{\mu}_i^{\text{cell}} + \boldsymbol{\epsilon} \odot \exp\left(0.5 \cdot \log(\boldsymbol{\sigma}_i^{\text{cell}})^2\right) \quad (1) \\
h_i^{\text{cell}} &= g_{\psi^{\text{cell}}}^{\text{cell}}(z_i^{\text{cell}}) \quad (2)
\end{aligned}
$$

where $\boldsymbol{\epsilon} \sim \mathcal{N}(\mathbf{0}, \boldsymbol{I})$ and $\odot$ denotes element-wise multiplication. $g_{\psi^{\text{cell}}}^{\text{cell}}(\cdot)$ represents the cell decoder without a conventional final linear transformation layer, and outputs $h_i^{\text{cell}} \in \mathbb{R}^l$ for cell $i$.

### 3.2.2. GENE ENCODER AND DECODER

Similarly, the gene encoder $f_{\phi^{\text{gene}}}^{\text{gene}}(\cdot)$ maps each gene $j$'s external static embedding $g_j \in \mathbb{R}^D$, derived from a pretrained LLM, into the contextual embedding space by predicting the parameters of a $K$-dimensional variational posterior distribution with mean $\boldsymbol{\mu}_j^{\text{gene}} \in \mathbb{R}^k$ and variance $(\boldsymbol{\sigma}_j^{\text{gene}})^2 \in \mathbb{R}^k$. The gene-level decoder $g_{\psi^{\text{gene}}}^{\text{gene}}(\cdot)$ is then used to produce output $h_j^{\text{gene}}$:

$$
\begin{aligned}
\left(\boldsymbol{\mu}_j^{\text{gene}}, \boldsymbol{\sigma}_j^{\text{gene}}\right) &\leftarrow f_{\phi^{\text{gene}}}^{\text{gene}}(g_j) \\
z_j^{\text{gene}} &= \boldsymbol{\mu}_j^{\text{gene}} + \boldsymbol{\epsilon} \odot \exp\left(0.5 \cdot \log(\boldsymbol{\sigma}_j^{\text{gene}})^2\right) \quad (3) \\
h_j^{\text{gene}} &= g_{\psi^{\text{gene}}}^{\text{gene}}(z_j^{\text{gene}}) \quad (4)
\end{aligned}
$$

### 3.2.3. SCILAMA RECONSTRUCTION OUTPUT

Similar to the siVAE framework (Choi et al., 2023), the output of sciLaMA is the reconstruction of the single cell expression data for gene $j$ in cell $i$, denoted as $\hat{c}_{i,j}$, via combining the respective cell and gene decoder outputs $h_i^{\text{cell}}$ and $h_j^{\text{gene}}$:

$$
\hat{c}_{i,j} = \left(h_i^{\text{cell}}\right)^T \times h_j^{\text{gene}} + b_j \quad (5)
$$

## 3.3. Optimization

The optimization of the sciLaMA framework involves a step-wise training procedure designed for representation learning of both cells and genes (Appendix B), and the training objectives follow the evidence lower bound (ELBO) framework, combining reconstruction accuracy and regularization via Kullback–Leibler (KL) divergence.

**Step 1: Pretraining the Cell Encoder and Decoder**: We first pretrain the weights of the cell encoder and decoder ($\phi^{\text{cell}}$ and $\psi^{\text{cell}}$, respectively) by treating the encoder-decoder as a VAE, where the objective function focuses on matching cell decoder outputs $h_i^{\text{cell}}$ to the original expression vectors $c_i$ via a linear transformation with parameters $\boldsymbol{W}^{\text{cell}} \in \mathbb{R}^{l \times M}$ and $\boldsymbol{b}$. The loss function $\mathcal{L}_{\text{cell}}$ for this step is defined as:

$$
\hat{c}_i^{\text{cellrecon}} = \left(h_i^{\text{cell}}\right)^T \times \boldsymbol{W}^{\text{cell}} + \boldsymbol{b} \quad (6)
$$

$$
\mathcal{L}_i^{\text{cellrecon}} = \left(c_i - \hat{c}_i^{\text{cellrecon}}\right)^T \left(c_i - \hat{c}_i^{\text{cellrecon}}\right) \quad (7)
$$

$$
\mathcal{L}_{\text{cell}} = \sum_i \mathcal{L}_i^{\text{cellrecon}} + \beta \cdot KL\left(\mathcal{N}(z_i^{\text{cell}}|\boldsymbol{\mu}_i^{\text{cell}}, \boldsymbol{\sigma}_i^{\text{cell}})\|\mathcal{N}(\mathbf{0}, \boldsymbol{I})\right) \quad (8)
$$

where $\beta$ represents the weight of the KL divergence term in VAEs, and is tuned to prioritize accurate reconstruction during the early stages of training.

**Step 2: Pretraining the Gene Encoder and Decoder**: Once the cell encoder and decoder are pretrained, its parameters ($\phi^{\text{cell}}$, $\psi^{\text{cell}}$, $\boldsymbol{W}^{\text{cell}}$, and $\boldsymbol{b}$) are frozen, and we then pretrain the parameters ($\phi^{\text{gene}}$, $\psi^{\text{gene}}$) of the gene encoder $f_{\phi^{\text{gene}}}^{\text{gene}}(\cdot)$ and decoder $g_{\psi^{\text{gene}}}^{\text{gene}}(\cdot)$, respectively. The loss function $\mathcal{L}_{\text{gene}}$ for this step is defined as:

$$
\mathcal{L}_i^{\text{recon}} = \left(c_i - \hat{c}_i\right)^T \left(c_i - \hat{c}_i\right) \quad (9)
$$

$$
\mathcal{L}_{\text{gene}} = \sum_i \mathcal{L}_i^{\text{recon}} + \beta \cdot KL\left(\mathcal{N}(z_j^{\text{gene}}|\boldsymbol{\mu}_j^{\text{gene}}, \boldsymbol{\sigma}_j^{\text{gene}})\|\mathcal{N}(\mathbf{0}, \boldsymbol{I})\right) \quad (10)
$$

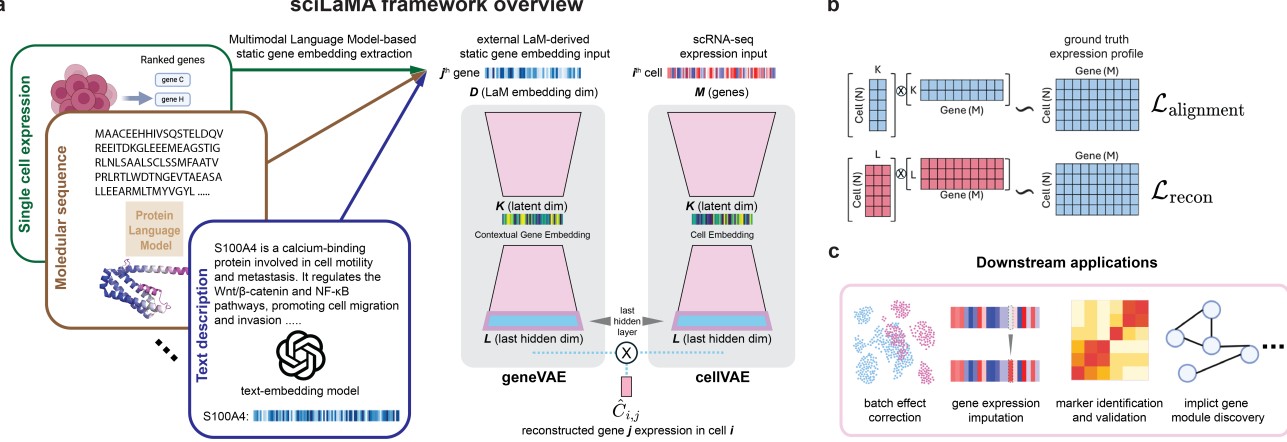

Figure 1. **sciLaMA overview**. **(a)** Diagram of the sciLaMA framework, which utilizes static gene embeddings generated from multimodal language models and employs paired encoder-decoders for both genes and cells. **(b)** Visualizations of cell and gene latent and last-hidden spaces and their operations for different components of the loss functions. **(c)** Illustrations of downstream applications using sciLaMA.

Note that unlike the reconstruction term $\mathcal{L}_i^{\text{cellrecon}}$ from previous step (Equation (7)), this loss function operates on the outputs of the last hidden layers of both cell and gene decoders (Equation (5)). Because the inputs to the gene encoder are the prior LLM-defined gene embeddings $\boldsymbol{g}_j$, and the output is reconstruction of the gene expression measurements $\boldsymbol{c}_i$, this pretraining serves to help adapt the LLM embeddings to the current (gene expression) context.

**Step 3: Joint Optimization of sciLaMA**: In the final step, all parameters of the sciLaMA framework are optimized to improve the reconstruction quality of the expression matrix. The loss function $\mathcal{L}_{\text{sciLaMA}}$ for this step is:

$$\hat{\boldsymbol{c}}_{i,j}^{\text{alignment}} = \left(\boldsymbol{z}_i^{\text{cell}}\right)^T \times \boldsymbol{z}_j^{\text{gene}} + b_j \tag{11}$$

$$\mathcal{L}_i^{\text{alignment}} = \left(\boldsymbol{c}_i - \hat{\boldsymbol{c}}_i^{\text{alignment}}\right)^T \left(\boldsymbol{c}_i - \hat{\boldsymbol{c}}_i^{\text{alignment}}\right) \tag{12}$$

$$\begin{aligned}
\mathcal{L}_{\text{sciLaMA}} = & \sum_i \mathcal{L}_i^{\text{recon}} + \gamma \cdot \mathcal{L}_i^{\text{alignment}} \\
& + \beta \cdot KL\left(\mathcal{N}(\boldsymbol{z}_i^{\text{cell}}|\boldsymbol{\mu}_i^{\text{cell}}, \boldsymbol{\sigma}_i^{\text{cell}})\|\mathcal{N}(\boldsymbol{0}, \boldsymbol{I})\right) \\
& + \beta \cdot KL\left(\mathcal{N}(\boldsymbol{z}_j^{\text{gene}}|\boldsymbol{\mu}_j^{\text{gene}}, \boldsymbol{\sigma}_j^{\text{gene}})\|\mathcal{N}(\boldsymbol{0}, \boldsymbol{I})\right)
\end{aligned} \tag{13}$$

where $\mathcal{L}_i^{\text{alignment}}$ is a reconstruction-based regularization term that encourages alignment between the latent spaces of cells ($\boldsymbol{z}_i^{\text{cell}}$) and genes ($\boldsymbol{z}_j^{\text{gene}}$) by enforcing that the linear product of the embeddings approximates the original expression value of gene $j$ in cell $i$ ($c_{i,j}$). This term, inspired by siVAE, serves as the interpretability term, ensuring that individual dimensions of the cell and gene embeddings ($\boldsymbol{z}^{\text{cell}}$ and $\boldsymbol{z}^{\text{gene}}$) correspond meaningfully to each other. $\gamma$ is a scalar weight (default = 0.05) that determines the influence of $\mathcal{L}_i^{\text{alignment}}$ term on the overall loss function. A small value prevents it from dominating the optimization process.

### 3.4. Inference and Embedding Extraction

After training the sciLaMA framework, the learned cell and gene embeddings can be extracted for downstream analyses. Given the trained encoders $f_{\hat{\phi}^{\text{cell}}}^{\text{cell}}(\cdot)$ and $f_{\hat{\phi}^{\text{gene}}}^{\text{gene}}(\cdot)$, they can be used to project a cell expression profile $\boldsymbol{c}^{(1)}$ or gene embedding $\boldsymbol{g}^{(2)}$ into the cell ($\boldsymbol{z}^{(1)}$) or gene ($\boldsymbol{z}^{(2)}$) latent space for downstream visualization or analysis.

$$(\boldsymbol{\mu}^{(1)}, \boldsymbol{\sigma}^{(1)}) \leftarrow f_{\hat{\phi}^{\text{cell}}}^{\text{cell}}(\boldsymbol{c}^{(1)}) \tag{14}$$

$$\boldsymbol{z}^{(1)} \sim \mathcal{N}\left(\boldsymbol{\mu}^{(1)}, \boldsymbol{\sigma}^{(1)}\right) \tag{15}$$

$$(\boldsymbol{\mu}^{(2)}, \boldsymbol{\sigma}^{(2)}) \leftarrow f_{\hat{\phi}^{\text{gene}}}^{\text{gene}}(\boldsymbol{g}^{(2)}) \tag{16}$$

$$\boldsymbol{z}^{(2)} \sim \mathcal{N}\left(\boldsymbol{\mu}^{(2)}, \boldsymbol{\sigma}^{(2)}\right) \tag{17}$$

## 4. Experiments

The experiments evaluating sciLaMA are designed to assess its performance in single-cell analysis at both cell- and gene-level tasks. For cell-level tasks, sciLaMA is assessed by evaluating its capacity to generate cell embeddings that simultaneously preserve biological signals and remove batch effects, with performance measured by (1) cell clustering annotation accuracy, (2) cell type separation precision, and (3) the effectiveness of batch mixing. For gene-level tasks, sciLaMA is evaluated on its ability to impute gene expression, identify gene markers, infer developmental trajectories and discover temporal dynamic gene modules (Figure 1**c**). Detailed methodologies are listed in the Appendices C and D.

### 4.1. Prior Knowledge Improves Cell Representation Learning

We first evaluated cell-level tasks because gene-level analysis tasks are largely cell state-specific, and therefore rely on

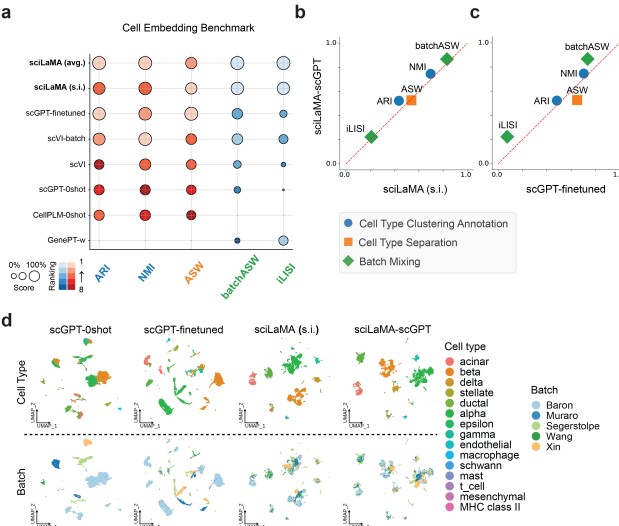

*Figure 2.* **Robust cell representation learning and integration with sciLaMA. (a)** Quantitative performance comparison of models based on sciLaMA against other methods in preserving biological variance (blue and orange metrics) and removing batch effects (green metrics). **(b-c)** Scatter plot directly comparing sciLaMA-GPT (y-axis) to sciLaMA (s.i.) (x-axis, **b**) and fine-tuned scGPT (x-axis, **c**). **(d)** UMAP (McInnes et al., 2020) visualizations of cell embeddings with colors indicating cell types (top) and batch origins (bottom).

*Table 1.* Cell representation learning and integration performance on human pancreatic datasets. Adjusted Rand Index (ARI) and Normalized Mutual Information (NMI) for cluster annotation accuracy; Average Silhouette Width (ASW) for cell type separation; batchASW and graph integration local inverse Simpson's Index (iLISI) for batch mixing quality.

| Methods | ARI ↑ | NMI ↑ | ASW ↑ | batchASW ↑ | iLISI ↑ |
|---|---|---|---|---|---|
| sciLaMA (avg.) | **0.522** | **0.745** | 0.535 | **0.865** | **0.238** |
| sciLaMA (s.i.) | 0.436 | 0.698 | 0.539 | 0.832 | 0.210 |
| scGPT fine-tuned | 0.483 | 0.704 | **0.650** | 0.736 | 0.074 |
| scVI-batch | 0.447 | 0.718 | 0.499 | 0.744 | 0.115 |
| scVI-raw | 0.297 | 0.570 | 0.453 | 0.621 | 0.030 |
| scGPT zero-shot | 0.321 | 0.487 | 0.442 | 0.588 | 0.005 |
| CellPLM zero-shot | 0.330 | 0.516 | 0.421 | 0.492 | $1.11e^{-16}$ |
| GenePT-w | 0.022 | 0.079 | 0.192 | 0.553 | 0.121 |

cell-level tasks such as accurate cell clustering and robust cell representations. To evaluate sciLaMA's performance and assess the impact of incorporating prior knowledge encoded as gene embeddings on cell-level tasks (Section 4), we benchmarked sciLaMA against the state-of-the-art (SOTA) model scVI (Lopez et al., 2018) and foundation models such as scGPT, CellPLM, and GenePT (Chen & Zou, 2024; Cui et al., 2024; Wen et al., 2023). Multiple variants of sciLaMA were created, each using a different set of gene embeddings precomputed using different prior knowledge databases to determine which prior knowledge is most relevant to single cell tasks: sciLaMA-GenePT, sciLaMA-ProtTrans, sciLaMA-CellPLM, sciLaMA-ChatGPT, and sciLaMA-ESM. To determine the extent to which the sciLaMA framework itself is superior to other models, we created the "self-informed" version of sciLaMA, sciLaMA (s.i.), to represent the framework when learning gene embeddings from the transposed single cell expression data itself solely (without prior LLM-derived knowledge). Cell-level tasks were evaluated using five pancreatic scRNA-seq datasets from different labs and sequencing platforms (Tran et al., 2020).

Across multiple standard integration metrics (Luecken et al., 2022), all sciLaMA variants robustly outperformed other models both individually (Figure 2**a**,**d**, Appendix E) as well as on average (Table 1), suggesting that the sciLaMA framework is a general, powerful framework for tackling cell-level tasks. For cell type clustering and annotation, sciLaMA achieved an average adjusted Rand index (ARI) of 0.522 and normalized mutual information (NMI) of 0.745, outperforming scVI (with batch variable consideration) by 16.78% and 3.76%, respectively, and fine-tuned scGPT by 8.07% and 5.82%. Additionally, its ARI and NMI values were approximately 1.5 times higher than those of the best zero-shot foundation models, showcasing its ability to generate well-defined cell clusters aligned with cell type annotations from the original studies. In cell type separation, sciLaMA achieved an average silhouette width (ASW) of 0.535 and a graph cell type local inverse Simpson's index (cLISI) of 0.9935, indicating precise separation of cell types with preserved biological variation. Furthermore, for batch effect correction, sciLaMA achieved the highest batch-ASW of 0.865 and a graph integration-LISI (iLISI) of 0.238, surpassing the next-best models by 16.26% and 96.69%, respectively. These results collectively highlight sciLaMA's robust ability to integrate cells across batches while maintaining accurate cell type representations and biological relevance.

Interestingly, the performance of sciLaMA (s.i.) without any external prior knowledge from LLMs is worse than all variants of sciLaMA with prior knowledge despite the diversity of prior knowledgebases, suggesting that incorporating prior knowledge of gene function is broadly acting to regularize sciLaMA and prevent overfitting (Figure 2**b**). These results are consistent with the observation across all tasks. sciLaMA outperformed scVI, another SOTA VAE-based model without external knowledge, again supporting that incorporating prior gene knowledge is beneficial to single cell analysis.

While our experiment above confirmed incorporating prior knowledge is helpful for single cell analysis, we also wondered whether with a framework inspired by paired VAEs, is sciLaMA the best framework for integrating prior knowledge? To explore this, we directly compared the transformer-based foundation model scGPT that was subsequently fine-

*Table 2.* Comparison of runtime (in seconds) for modeling 14,767 human pancreatic cells sourced from five different origins on a single NVIDIA A100 80GB GPU. Due to memory limitations, the batch size for scGPT was set to 10, while siVAE and the various sciLaMA configurations utilized a batch size of 128.

| Method | scGPT fine-tune | siVAE | sciLaMA (avg.) |
|---|---|---|---|
| **Runtime (s)** ↓ | 19,474 | 2,265 | **759** |

tuned on our training single cell data (scGPT-finetuned) with sciLaMA-scGPT (sciLaMA using pretrained scGPT gene embeddings). Both models are based on the same pretrained scGPT-whole-human as prior knowledge, but differ in how the pretrained embeddings are updated further. sciLaMA-scGPT outperformed fine-tuned scGPT by 6.82% in cell type clustering and annotation task (ARI and NMI) (Figure 2**c**). Although the fine-tuned scGPT achieved marginally better results in silhouette width (ASW), its lower batch-ASW and integration-LISI (iLISI) scores (by 34.57% on average) indicate poor batch integration. This comparison underscores the lightweight and well-designed nature of sciLaMA, which improves performance while being more computationally efficient, reducing runtime by 25-fold compared to fine-tuned scGPT (Table 2).

### 4.2. sciLaMA Reconstructs Gene Expression with High Accuracy

We next benchmarked sciLaMA accuracy on gene-level tasks, starting with the imputation of gene expression patterns. Gene imputation, the prediction of missing or masked gene expression levels based on other genes' profiles, is particularly beneficial for sparsely measured datasets, such as Multiplexed Error-Robust Fluorescence in situ Hybridization (MERFISH) or Antibody-Derived Tags (ADTs), where only a subset of genes is typically quantified in an experiment. We benchmarked sciLaMA against leading models for gene imputation accuracy, including scProjection, gimVI, uniPort and Tangram (Johansen et al., 2023; Lopez et al., 2019; Cao et al., 2022; Biancalani et al., 2021). The experimental setup employed a leave-one-gene-out strategy, where the expression of a single gene was masked across all cells, and the models were tasked with predicting its expression pattern based on the remaining genes.

Our results show that sciLaMA models consistently outperformed competing models in imputation accuracy (Figure 3**a,b**, and Table 3) on the spatial transcriptomics data (Codeluppi et al., 2018). sciLaMA achieved the highest scores across established metrics (Appendix D) (Li et al., 2022), outperforming the average performance of other benchmarked methods (Johansen et al., 2023; Lopez et al., 2019; Cao et al., 2022; Biancalani et al., 2021) by 27.39% in Pearson Correlation Coefficient (PCC), 15.58% in Spearman Correlation Coefficient (SCC), 32.86% in 1-

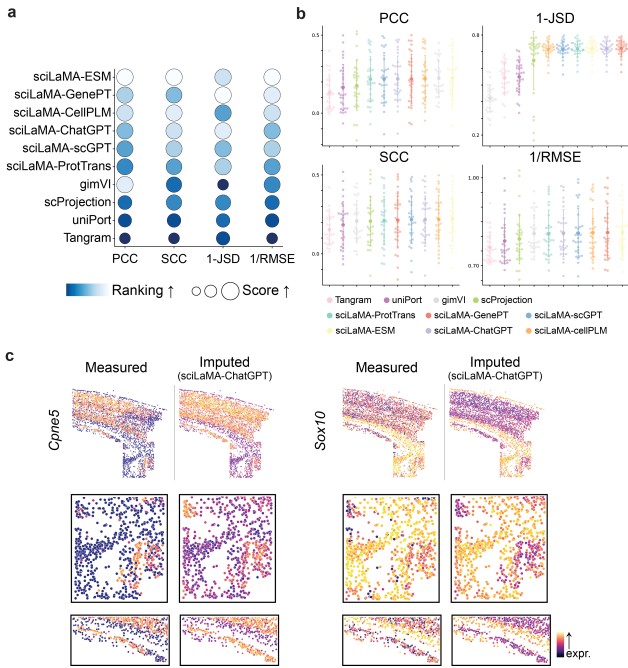

*Figure 3.* **Accurate imputation of unseen gene expression with sciLaMA. (a)** Quantitative performance comparison of models based on sciLaMA against other methods for gene imputation task using leave-one-gene-out strategy. (**b**) Metric values for 30 genes from the spatial dataset across methods (color-coded). (**c**) Example visualizations of measured (left) and imputed (right) spatial gene expression patterns.

Jensen–Shannon Divergence (1-JSD), and 3.32% in 1/Root Mean Squared Error (1/RMSE) on average. These metrics indicate that its predictions were more aligned with true gene expression patterns compared to other models (Figure 3**a**). Notably, the results demonstrate the significance of incorporating external gene information gain, as evidenced by sciLaMA's performance superiority over the baseline sciLaMA (s.i.) model, as well as additional baseline sciLaMA (random) and sciLaMA (shuffled) (Appendix E). Unlike sciLaMA, these baseline models do not leverage meaningful prior knowledge derived from LLMs. Specifically, sciLaMA (s.i.) utilizes a transposed single-cell expression matrix, sciLaMA (random) employs a randomized input matrix for the gene encoder, and sciLaMA (shuffled) uses shuffled external gene embeddings to intentionally disrupt dimension alignment. Collectively, these comparisons emphasize the significance of leveraging structured, semantically meaningful gene embeddings derived from LLMs to enhance generalizability.

Figure 3**c** illustrates examples of measured versus imputed spatial patterns for genes such as *Cpne5* and *Sox10* and show sciLaMA accurately predicts expression while preserving spatial organization and region-specific heterogeneity of expression, which is crucial for understanding tissue spatial

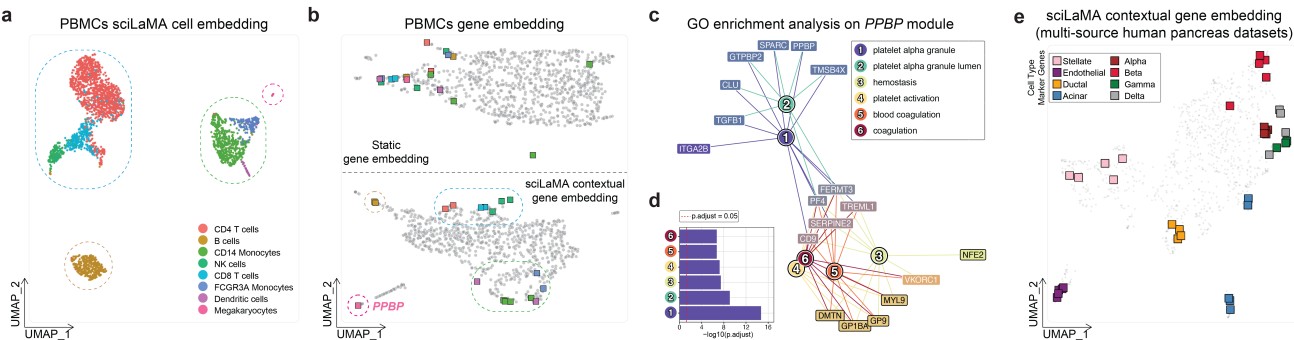

*Figure 4.* **Marker gene identification and validation using sciLaMA. (a)** UMAP of human PBMC 3K dataset cell embedding using sciLaMA, with points representing cells colored by cell type and outlined by coarse cell classes using dashed circles. **(b)** Comparison of LLM-derived static gene embedding (top) and sciLaMA-derived contextual gene embedding (bottom) with points representing genes. Marker genes are colored by cell type specificity, and those from the same circle are relevant to the same broader cell classes. Color codes are consistent between (a) and (b). **(c)** A graph of a gene module identified through sciLaMA-based gene clustering, with Gene Ontology (GO) terms enriched for module-associated genes. The module includes PPBP gene, a known marker for Megakaryocytes. **(d)** Bar chat of the top six GO terms and significance (adjusted p-values). **(e)** UMAP visualization of sciLaMA contextual gene embedding on multi-source human pancreas datasets. Marker gene modules associated with different cell types are highlighted.

*Table 3.* Evaluation of gene expression imputation performance on spatial transcriptomics data across multiple methods using Pearson Correlation Coefficient (PCC), Spearman Correlation Coefficient (SCC), Jensen-Shannon Divergence (JSD), and Root Mean Square Error (RMSE). A leave-one-gene-out strategy was applied on 30 measured genes.

| Methods | PCC (↑) | SCC (↑) | JSD (↓) | RMSE (↓) |
|---|---|---|---|---|
| sciLaMA (avg.) | 0.222 ± 0.027 | **0.217 ± 0.028** | **0.283 ± 0.008** | **1.242 ± 0.022** |
| scProjection | 0.177 ± 0.029 | 0.207 ± 0.029 | 0.352 ± 0.032 | 1.277 ± 0.023 |
| gimVI | **0.224 ± 0.021** | 0.207 ± 0.024 | 0.580 ± 0.014 | 1.243 ± 0.017 |
| uniPort | 0.166 ± 0.027 | 0.184 ± 0.027 | 0.451 ± 0.017 | 1.287 ± 0.022 |
| Tangram | 0.130 ± 0.019 | 0.154 ± 0.018 | 0.458 ± 0.017 | 1.316 ± 0.015 |

structure.

### 4.3. sciLaMA Enables Marker Gene Identification

In single-cell studies, identifying and validating marker genes characteristic of individual cell types is another essential process for cell type annotation traditionally dependent on expert domain knowledge. Conventionally, bioinformaticians preprocess and integrate data, cluster cells, and then experts annotate these clusters using known biomarkers or gene signatures relevant to specific cell types (Butler et al., 2018; Wolf et al., 2018). Such division of labor is time-consuming and demands extensive collaboration. sciLaMA streamlines this process by simultaneously integrating cells and implicitly organizing genes into biologically meaningful modules within its contextual gene representation space. By analyzing gene embeddings, sciLaMA can identify groups of genes that are consistently co-expressed or show coordinated patterns within specific cell types. This goes beyond simply checking the expression levels of predefined markers such as CD4 for T-cells. Instead, it reveals potentially unknown gene modules that strongly correlate

with particular cellular states or types. sciLaMA not only reduces the manual labor involved in marker identification but also opens up possibilities for discovering new biological insights by detecting subtle, coordinated gene expression patterns that expert-driven methods might overlook.

To assess sciLaMA's efficacy in marker gene identification, we compared its contextual gene embeddings to static embeddings from the LLMs. sciLaMA's contextualization significantly improved the clustering of markers associated with the same cell states (Figure 4**a-b**). For example, in the static embeddings (Figure 4**b**, top), marker genes for the same cell type do not cluster as expected, while in the sciLaMA contextual embeddings (Figure 4**b**, bottom), markers for the same cell states group together, as indicated by the circles. Moreover, *PPBP* is a well-established marker for Megakaryocytes (platelet precursor cells) in human peripheral blood mononuclear cells (PBMCs) (Butler et al., 2018), and sciLaMA's contextual gene embedding presents a cluster that includes it. Neighboring genes within this cluster were linked to platelet-related biological processes, cellular components, and molecular functions, confirmed via Gene Ontology (GO) enrichment analysis (Figure 4**c-d**) (Subramanian et al., 2005). Many of these genes, though not previously annotated as Megakaryocyte markers from the original study, exhibit strong co-expression with PPBP and functional links to platelet biology. Their coordinated clustering in biologically meaningful modules indicates their relevance to Megakaryocyte identity.

Furthermore, sciLaMA robustly identified marker modules across multiple datasets, demonstrating its effectiveness even in the presence of batch effects (Figure 4**e**). Importantly, sciLaMA integrating LLM-derived priors gene knowledge outperformed the self-informed version sciL-

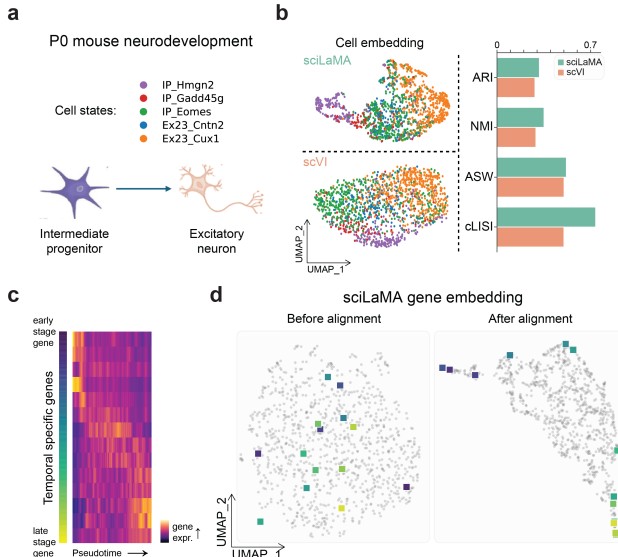

*Figure 5.* **Enhanced developmental gene trajectory analysis with sciLaMA.** **(a)** Overview of P0 mouse neurodevelopment data, with five cell types from early progenitors to mature excitatory neurons. **(b)** UMAP visualizations of cell embeddings using sciLaMA (top) and scVI (bottom) with a bar plot comparing cell type annotation and separation performance. **(c)** Pseudotime (x-axis) heatmap displaying the dynamic changes in gene expression across developmental stages. Rows represent ordered temporal specific genes. **(d)** UMAP visualizations of gene embeddings without (left) and with (right) embedding alignment using sciLaMA. Temporal specific genes (from (c)) are highlighted with color gradient.

aMA (s.i.) across clustering metrics (Table S7), which indicates the value of leveraging pretrained static gene embeddings. These findings highlight sciLaMA's potential to streamline single-cell studies by reducing reliance on manual annotation and revealing novel biological insights, which advances gene module discovery.

## 4.4. sciLaMA Enhances Trajectory Analysis by Unveiling Temporal Dynamics of Genes

Building upon its strength in identifying gene markers and modules across discrete cell types, sciLaMA also excels at capturing temporal dynamics in developmental processes. This capability enables the study of continuous gene expression changes across time and facilitates the analysis of cell differentiation and developmental trajectories.

To investigate sciLaMA's capability in this context, we conducted pseudotime trajectory analysis using cell embeddings learned by sciLaMA and compared them with those from scVI, a SOTA single-cell model. The analysis was applied to a dataset capturing P0 mouse cortex development (Figure 5a) (Chen et al., 2019). Pseudotime visualizations (Figure 5b, and Appendix E) illustrated that sciLaMA provided

*Table 4.* Cell representation learning performance on P0 mouse neurodevelopment dataset, with ARI and NMI quantifying cluster annotation accuracy, and ASW and cLISI quantifying cell type separation.

| Methods | ARI ↑ | NMI ↑ | ASW ↑ | cLISI ↑ |
|---------|-------|-------|-------|---------|
| sciLaMA | **0.316** | **0.351** | **0.518** | **0.738** |
| scVI | 0.284 | 0.291 | 0.501 | 0.501 |

clearer transitions between developmental stages, such as the progression from intermediate progenitors (IPs) to layer-2-3 excitatory neurons (ExNs). sciLaMA outperformed scVI in trajectory clarity by 20.65% overall (Table 4).

Pseudotime-aligned heatmaps of gene expression (Figure 5c, and Appendix E) highlighted temporal-specific genes with coordinated expression shifts corresponding to distinct stages of cell differentiation. Additionally, sciLaMA's contextual gene embeddings further illuminated temporal relationships between genes, offering insights into the sequential activation of developmental markers (Figure 5d). This analysis provides a comprehensive perspective on the dynamic interplay of genes during cell differentiation and development.

By accurately mapping cell lineages and identifying stage-specific gene modules, sciLaMA provides researchers with a powerful tool for understanding cell differentiation and developmental processes. When applied to organoid datasets, sciLaMA can also compare developmental trajectories of organoids with those of real tissues. For example, it can identify which gene modules from real tissues correspond to specific stages in organoid development, aiding in the assessment of organoid fidelity. This capability has significant implications for therapeutic strategies, enabling researchers to evaluate how organoids can model human diseases and inform potential treatment designs.

## 5. Conclusion

This study introduces sciLaMA, a novel framework that integrates external gene knowledge from language models with single-cell expression data to address critical challenges in single-cell analysis and enable comprehensive downstream tasks spanning both cell-level and gene-level analyses. Our experiments demonstrate the framework's effectiveness and performance superiority, and highlight the value of incorporating external gene knowledge through an innovative design. These findings establish sciLaMA as a powerful tool for advancing our understanding of cellular heterogeneity and gene regulation, and showcase how language models can be leveraged through a lightweight adapter framework.

## Acknowledgements

This work began during H.H.'s research internship at Microsoft Research and was supported by an NSF CAREER award (1846559, G.Q.). Additional funding came from the National Institutes of Health, including the Office of the Director/National Institute of Mental Health (DP2 MH129987, G.Q.) and the National Institute of Child Health and Human Development (P50 HD103526). We thank Erdal Cosgun, Shuangjia Lu, and the Microsoft Research Health Futures team for their guidance and technical insights.

## Impact Statement

This paper aims to integrate contextual and general knowledge about cells and genes to enhance single-cell data analysis. By leveraging large language models from different modalities, our methodology supports critical tasks with improved performance and computational efficiency. This approach has the potential to advance the field of Machine Learning in scientific research by facilitating the study of cellular heterogeneity, gene regulation, and developmental processes across various biological contexts. Furthermore, its ability to streamline analysis reduces reliance on extensive manual annotation and switching between disparate tools on various tasks. The implications of this approach include its contribution to biomedical research, where it may support therapeutic discovery and precision medicine. There are many potential societal consequences of our work, none of which we feel must be specifically highlighted here.

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

# Appendix
# sciLaMA: A Single-Cell Representation Learning Framework to Leverage Prior Knowledge from Large Language Models

## A. Model Input Processing:

### A.1. Cell Encoder Input

As mentioned in **Section 3 Methods**, the input for cell Encoder is the scRNA-seq data for specific cell population $c$. Therefore, $c_{i,j}$ denotes the scaled log-normalized expression value of gene $j$ in cell $i$.

The raw scRNA-seq expression matrix, $c^{\text{raw}}$, is a sparse count matrix. For use in sciLaMA, the data after quality control (QC) is processed through library size normalization and feature-wise z-score scaling to achieve zero mean and unit variance. Values beyond ±10 are clipped. Specifically, the normalized expression $c^{\text{norm}}$ is calculated as:

$$c_{i,j}^{\text{norm}} = \log_e \left( 1 + 10^4 \times \frac{c_{i,j}^{\text{raw}}}{\sum_{k=1}^{m} c_{i,k}^{\text{raw}}} \right)$$

Here, $c_{i,j}^{\text{raw}}$ represents the raw count value of gene $j$ in cell $i$, and $\sum_{k=1}^{m} c_{i,k}^{\text{raw}}$ is the total expression counts number for cell $i$. The multiplication by $10^4$ ensures a standardized size factor for normalization. This normalization procedure adjusts for library size differences across cells and prepares the data for following analysis.

### A.2. Gene Encoder Input

In this study, sciLaMA integrated static gene embeddings from six external sources across three distinct modalities.

| Source | Dimensionality | Modality |
|---|---|---|
| ChatGPT | 1536 | Text |
| GenePT (NCBI) | 1536 | Text |
| ESM | 5120 | Protein Sequence |
| ProtTrans | 1024 | Protein Sequence |
| scGPT | 512 | Single Cell |
| CellPLM | 1024 | Single Cell |

*Table S1.* Gene Embedding Sources and Characteristics.

#### A.2.1 Natural Language Embeddings

We acquired text description-based gene embeddings from two studies: GenePT and scELMo (Chen & Zou, 2024; Liu et al., 2023), utilizing the OpenAI text-embedding-ada-002 model (OpenAI, 2022). These embeddings were generated using two distinct text corpora: GPT-3.5 generated descriptions (referred to as ChatGPT) and National Center for Biotechnology Information (NCBI) gene card summaries (referred to as GenePT). We obtained 1,536-dimensional static embeddings for each gene ($d = 1,536$).

#### A.2.2 Protein Language Embeddings

We derived protein sequence-based gene embeddings from two protein language models: ESM2 t48_15B_UR50D with 5,120-dimensional embeddings per gene (Lin et al., 2023), and ProtXLNet from ProtTrans with 1,024-dimensional embeddings (Elnaggar et al., 2022) from the SATURN study (Rosen et al., 2024). These embeddings were generated using the amino acid sequences of each corresponding gene.

#### A.2.3 Single-Cell Gene Language Embeddings

For single-cell foundation models, we retrieved static gene embeddings from two pretrained models: scGPT-whole-human (512-dimensional embeddings) (Cui et al., 2024) and cellPLM (1,024-dimensional embeddings) (Wen et al., 2023). The scGPT embeddings were obtained using the model's GitHub tutorial, while cellPLM embeddings were extracted from the embedder module's feature encoder parameters, as directed by the authors.

## B. Model Optimization Illustration:

The sciLaMA model optimization process, comprehensively described in **Section 3 Methods**, is illustrated through a stepwise training strategy visualization (Figure S1). The optimal hyperparameter values chosen for our experiments are scalar $\gamma = 0.05$ and latent dimensionality $K = 40$, based on the evaluations presented in Table S8 and Table S9.

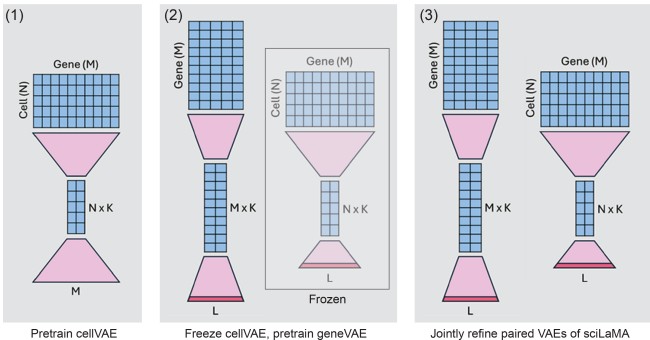

*Figure S1.* Schematic representation of the progressive optimization workflow for the sciLaMA framework. (Box indicates freezing parameters.)

## C. Dataset Introduction:

### C.1. Experiment 1: Cell Representation Learning Benchmarking

This experiment benchmarks cell representation learning methods using a combination of single-cell RNA sequencing datasets derived from five studies focused on the pancreas. The data includes a total of 14,767 cells spanning 13,062 genes (after intersection with precomputed static gene embeddings).

**Datasets Used:** Baron et al.: 8,569 cells (Baron et al., 2016); Segerstolpe et al.: 2,127 cells (Segerstolpe et al., 2016); Muraro et al.: 2,122 cells (Muraro et al., 2016); Xin et al.: 1,492 cells (Xin et al., 2016); Wang et al.: 457 cells (Wang et al., 2016).

The aggregated dataset was a gold-standard benchmarking dataset originally analyzed in the context of batch-effect correction, as described in Tran et al., in 2020 (Tran et al., 2020). This benchmarking experiment evaluates the performance of cell representation learning in mitigating batch effects while preserving biological signal.

### C.2. Experiment 2: Gene Expression Imputation Benchmarking

This experiment evaluates the accuracy of gene expression imputation approaches by leveraging two complementary datasets (Zeisel et al., 2015; Codeluppi et al., 2018):

**Reference scRNA-seq Dataset (Zeisel et al., 2015):**

· Number of cells: 3,005

· Total genes: 19,972, with 3,654 highly variable genes selected for benchmarking.

· Validation: A 10% validation split is used for early stopping during model training.

**Spatial Transcriptomics Dataset (Codeluppi et al., 2018):**

· Number of spatial spots: 4,530

· Genes: 30, analyzed using a leave-one-gene-out approach to simulate imputation scenarios.

This setup allows for assessing the generalizability of gene imputation models.

### C.3. Experiment 3: Marker Gene Identification

This experiment focuses on identifying marker genes for distinct cell types using the human Peripheral Blood Mononuclear Cell (PBMC) 3K dataset from 10x Genomics, a legacy dataset widely utilized in tools like Seurat (Butler et al., 2018) and scanpy (Wolf et al., 2018) tutorials. The ground truth gene markers and cell type annotations were obtained from the tutorials (`https://satijalab.org/seurat/articles/pbmc3k_tutorial`, and `https://scanpy.readthedocs.io/en/stable/tutorials/basics/clustering-2017.html`).

**Dataset Details:**

· Initial Size: 2,700 cells × 32,738 genes

· Final Size: 2,700 cells × 9,540 genes (post-filtering and intersection with static gene embeddings).

| Cell Type | Number of cells |
|---|---|
| CD4 T cells | 1158 |
| CD14 Monocytes | 487 |
| B cells | 357 |
| CD8 T cells | 329 |
| FCGR3A Monocytes | 160 |
| NK cells | 160 |
| Dendritic cells | 36 |
| Megakaryocytes | 13 |

*Table S2.* Cell Type Statistics of human PBMC 3K dataset

### C.4. Experiment 4: Trajectory Analysis and Temporal Dynamic Gene Discovery

This experiment investigates gene dynamics along developmental trajectories using the P0 mouse cortex dataset from the SNARE-seq study (Chen et al., 2019). The original SNARE-seq dataset includes both transcriptomic and epigenomic information from the same single cells, but we only utilized the transcriptomic data with 1,469 cells and 8,293 genes (after intersection with precomputed static gene embeddings). This experiment focuses on uncovering temporally dynamic genes critical for neurodevelopmental processes. The ground truth gene markers and cell type annotations were obtained from the original study.

| Cell Type | Number of cells |
|---|---|
| IP_Hmgn2 | 214 |
| IP_Gadd45g | 99 |
| IP_Eomes | 437 |
| Ex23_Cntn2 | 177 |
| Ex23_Cux1 | 542 |

*Table S3.* Cell Type Statistics of mouse P0 cortex dataset

## D. Benchmarking Metrics Introduction:

To comprehensively evaluate the performance of various methods, we employ metrics tailored to different aspects of single-cell data analysis, including cluster annotation accuracy, cell type separation, batch mixing quality, and predictive/imputation accuracy (Li et al., 2022; Luecken et al., 2022), briefly summarized below:

### D.1. Clustering and Annotation Accuracy

To assess the biological relevance of clustering and annotation based on the learned embeddings, we employ:

· **Adjusted Rand Index (ARI):** Measures the agreement between predicted and ground-truth cluster labels, adjusted for chance. A higher ARI indicates better alignment between predicted clusters and original biological annotations, reflecting more accurate and biologically meaningful clustering.

· **Normalized Mutual Information (NMI):** Quantifies the mutual dependence between predicted clusters and ground-truth cell type annotation labels, normalized to account for the total number of clusters. A higher NMI indicates better clustering accuracy.

### D.2. Cell Type Separation

To evaluate how well methods preserve separation between distinct cell types, we employ:

· **Average Silhouette Width (ASW):** Evaluates the cohesion within clusters and the separation between them. Higher ASW scores indicate that cells within the same cluster are more similar to each other than to cells in other clusters, signifying well-defined clusters.

· **Graph Cell-Type Integration Local Inverse Simpson's Index (cLISI):** Measures the local diversity of cell types within neighborhoods in an integrated graph representation. High cLISI values suggest better grouping of similar cell types in the embedding space.

### D.3. Batch-Effect Correction Quality

To evaluate batch effect removal while preserving biological variance, we apply:

· **Batch-Adjusted Silhouette Width (batchASW):** Evaluates the extent of batch mixing while penalizing over-mixing of unrelated cells. Higher batchASW scores indicate better batch integration without compromising biological separation.

· **Graph Integration Local Inverse Simpson's Index (iLISI):** Measures the diversity of batch labels within local neighborhoods of an integrated graph. Higher iLISI scores indicate more uniform batch mixing, reflecting better integration while preserving cell type integrity.

### D.4. Predictive Accuracy and Divergence Metrics

For imputation and gene expression prediction tasks, we employ:

· **Pearson Correlation Coefficient (PCC):** Assesses linear relationships between predicted and observed gene expression values, with higher values indicating stronger correlations.

· **Spearman Correlation Coefficient (SCC):** Evaluates rank-based relationships, capturing monotonic correlations between predicted and observed values, providing insights into the consistency of expression patterns.

· **Jensen-Shannon Divergence (JSD):** Measures the similarity between predicted and true gene expression distributions. Lower JSD values indicate better agreement between the two distributions.

· **Root Mean Square Error (RMSE):** Quantifies the average magnitude of errors between predicted and observed values. Lower RMSE scores reflect higher accuracy

### D.5. Clustering Quality Metrics

To evaluate the geometric coherence and separation of clusters in the learned gene embedding space, we include two additional metrics:

· **Davies-Bouldin Index (DBI):** Quantifies the ratio of intra-cluster dispersion to inter-cluster separation. Lower DBI values indicate better-defined clusters with high intra-cluster similarity and distinct separation between clusters.

· **Calinski-Harabasz Score (CHS):** Measures the ratio of between-cluster dispersion to within-cluster dispersion. Higher CHS values reflect dense, well-separated clusters

# E. Supplementary Results:

| Methods | w\ external knowledge | ARI ↑ | NMI ↑ | ASW ↑ | batchASW ↑ | iLISI ↑ |
|---|---|---|---|---|---|---|
| sciLaMA-GenePT | √ | 0.545 | 0.767 | 0.539 | 0.863 | 0.240 |
| sciLaMA-CellPLM | √ | 0.479 | 0.723 | 0.541 | 0.871 | 0.257 |
| sciLaMA-ProtTrans | √ | 0.547 | 0.749 | 0.538 | 0.864 | 0.229 |
| sciLaMA-ChatGPT | √ | 0.545 | 0.762 | 0.534 | 0.863 | 0.225 |
| sciLaMA-scGPT | √ | 0.522 | 0.746 | 0.526 | 0.867 | 0.223 |
| sciLaMA-ESM | √ | 0.494 | 0.722 | 0.529 | 0.864 | 0.253 |
| sciLaMA (s.i.) | × | 0.436 | 0.698 | 0.539 | 0.832 | 0.210 |

*Table S4.* Cell representation learning and integration performance on human pancreatic datasets across variants of sciLaMA models

| Methods | cLISI ↑ |
|---|---|
| sciLaMA-GenePT | 0.995 |
| sciLaMA-CellPLM | 0.995 |
| sciLaMA-ProtTrans | 0.992 |
| sciLaMA-ChatGPT | 0.993 |
| sciLaMA- scGPT | 0.993 |
| sciLaMA- ESM | 0.993 |
| sciLaMA (s.i.) | 0.987 |
| scGPT fine-tuned | 0.998 |
| scVI-batch | 0.982 |
| scVI-raw | 0.972 |
| scGPT zero-shot | 0.951 |
| CellPLM zero-shot | 0.961 |
| GenePT-w | 0.838 |

*Table S5.* Graph Cell-Type Integration Local Inverse Simpson's Index (cLISI) scores across methods (listed as supplementary result due to the low variance of 0.001714)

| Methods | w\ external knowledge | PCC (↑) | SCC (↑) | JSD (↓) | RMSE (↓) |
|---|---|---|---|---|---|
| sciLaMA-GenePT | √ | 0.220 ± 0.029 | 0.214 ± 0.031 | 0.280 ± 0.009 | 1.243 ± 0.023 |
| sciLaMA-CellPLM | √ | 0.222 ± 0.027 | 0.218 ± 0.028 | 0.286 ± 0.009 | 1.242 ± 0.022 |
| sciLaMA-ProtTrans | √ | 0.218 ± 0.026 | 0.211 ± 0.028 | 0.283 ± 0.009 | 1.246 ± 0.021 |
| sciLaMA-ChatGPT | √ | 0.219 ± 0.027 | 0.217 ± 0.027 | 0.282 ± 0.009 | 1.244 ± 0.022 |
| sciLaMA-scGPT | √ | 0.219 ± 0.027 | 0.217 ± 0.027 | 0.285 ± 0.009 | 1.244 ± 0.022 |
| sciLaMA-ESM | √ | 0.233 ± 0.026 | 0.227 ± 0.027 | 0.282 ± 0.009 | 1.233 ± 0.022 |
| sciLaMA (s.i.) | × | 0.202 ± 0.027 | 0.212 ± 0.025 | 0.286 ± 0.009 | 1.258 ± 0.022 |
| sciLaMA (random) | × | 0.051 ± 0.027 | 0.049 ± 0.031 | 0.289 ± 0.009 | 1.374 ± 0.020 |
| sciLaMA (shuffled) | × | 0.056 ± 0.036 | 0.043 ± 0.037 | 0.288 ± 0.009 | 1.366 ± 0.027 |

*Table S6.* Evaluation of gene expression imputation performance on spatial transcriptomics data across variants of sciLaMA models

| Methods | w/ external knowledge | Davies-Bouldin Index (↓) | Calinski-Harabasz Score (↑) |
|---|---|---|---|
| sciLaMA-GenePT | √ | 0.852 | 16.376 |
| sciLaMA-CellPLM | √ | 0.727 | 19.610 |
| sciLaMA-ProtTrans | √ | 0.802 | 19.947 |
| sciLaMA-ChatGPT | √ | 0.874 | 16.522 |
| sciLaMA-scGPT | √ | 0.780 | 17.973 |
| sciLaMA-ESM | √ | 0.780 | 16.920 |
| sciLaMA (s.i.) | × | 0.977 | 13.087 |

*Table S7.* Clustering performance comparison for marker gene identification across variants of sciLaMA models

| $\gamma$ | ARI mean | ARI std | NMI mean | NMI std | ASW mean | ASW std | cLISI mean | cLISI std |
|---|---|---|---|---|---|---|---|---|
| 0 | 0.464 | 0.371 | 0.513 | 0.390 | 0.589 | 0.087 | 0.881 | 0.170 |
| 0.01 | 0.582 | 0.021 | 0.734 | 0.022 | 0.645 | 0.010 | 0.990 | 0.002 |
| 0.05 | 0.665 | 0.114 | 0.763 | 0.041 | 0.654 | 0.005 | 0.991 | 0.002 |
| 0.1 | 0.634 | 0.088 | 0.751 | 0.023 | 0.658 | 0.012 | 0.992 | 0.002 |
| 0.25 | 0.581 | 0.024 | 0.743 | 0.013 | 0.655 | 0.006 | 0.990 | 0.002 |
| 0.5 | 0.592 | 0.010 | 0.747 | 0.004 | 0.658 | 0.012 | 0.993 | 0.001 |
| 0.75 | 0.590 | 0.015 | 0.748 | 0.013 | 0.656 | 0.012 | 0.992 | 0.002 |
| 1 | 0.647 | 0.107 | 0.762 | 0.038 | 0.651 | 0.025 | 0.993 | 0.003 |

*Table S8.* Effect of scalar $\gamma$ on clustering performance across multiple metrics.

| $K$ Latent dim | ARI mean | ARI std | NMI mean | NMI std | ASW mean | ASW std | cLISI mean | cLISI std |
|---|---|---|---|---|---|---|---|---|
| 10 | 0.651 | 0.110 | 0.756 | 0.042 | 0.654 | 0.005 | 0.991 | 0.002 |
| 20 | 0.627 | 0.082 | 0.761 | 0.025 | 0.631 | 0.015 | 0.991 | 0.000 |
| 30 | 0.583 | 0.017 | 0.742 | 0.022 | 0.633 | 0.014 | 0.991 | 0.001 |
| 40 | 0.680 | 0.114 | 0.771 | 0.039 | 0.637 | 0.009 | 0.991 | 0.002 |
| 50 | 0.606 | 0.085 | 0.743 | 0.023 | 0.631 | 0.010 | 0.990 | 0.002 |
| 60 | 0.649 | 0.100 | 0.757 | 0.030 | 0.631 | 0.015 | 0.990 | 0.002 |
| 70 | 0.590 | 0.037 | 0.738 | 0.017 | 0.635 | 0.009 | 0.991 | 0.001 |
| 80 | 0.649 | 0.095 | 0.753 | 0.020 | 0.632 | 0.009 | 0.991 | 0.001 |
| 90 | 0.656 | 0.102 | 0.756 | 0.030 | 0.637 | 0.008 | 0.991 | 0.001 |
| 100 | 0.641 | 0.108 | 0.751 | 0.027 | 0.635 | 0.006 | 0.991 | 0.002 |

*Table S9.* Effect of varying latent dimension $K$ on clustering performance across multiple metrics.

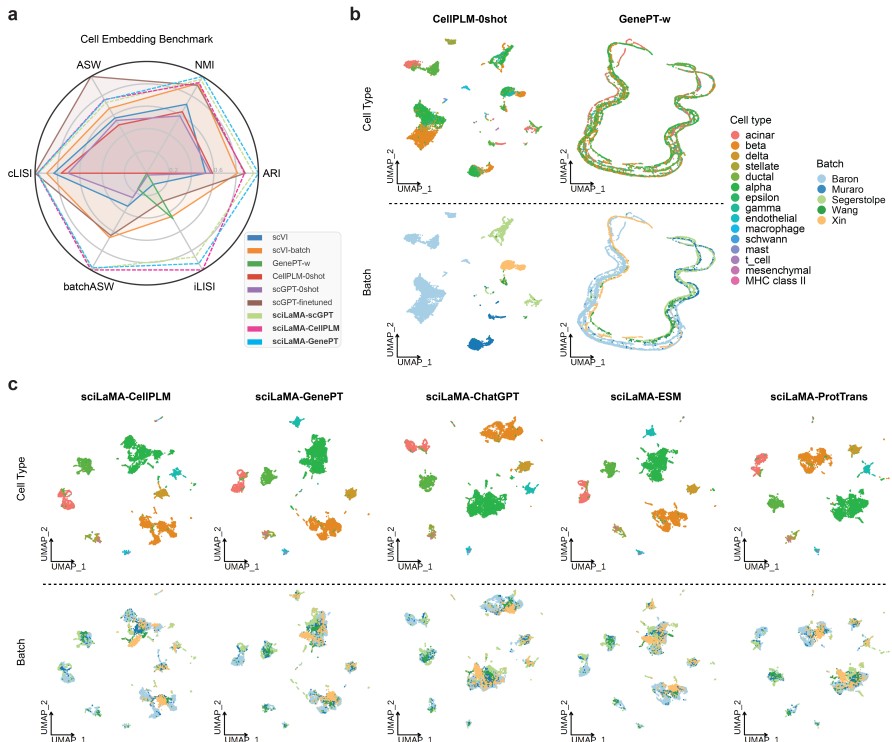

*Figure S2.* **Benchmark of cell representation learning. (a)** Radar plot showing the performance across six established metrics, comparing single cell SOTA methods (scVI and scVI-batch), zero-shot models (GenePT-w, CellPLM, and scGPT), a fine-tuned model (scGPT), and comparable sciLaMA-based models (sciLaMA-GenePT/CellPLM/scGPT). **(b-c)** UMAP visualizations of cell embeddings derived from various models, with colors indicating cell types (top) and batch origins (bottom). (b) includes foundation models in zero-shot mode, while (c) presents sciLaMA-based models in additional to those from Figure 2c.

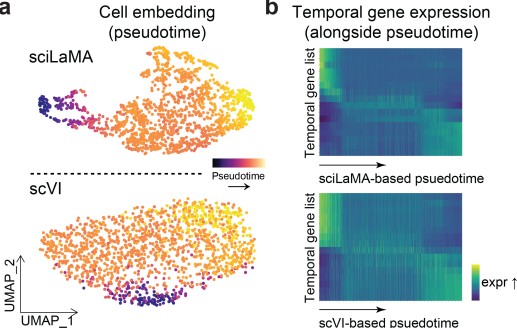

*Figure S3.* **Enhanced developmental cell trajectory analysis with sciLaMA. (a)** UMAP visualizations of cell embeddings from sciLaMA (top) and scVI (bottom) colored by inferred pseudotime via Palantir. **(b)** Heatmaps of dynamic gene expression changes by pseudotime (x-axis) with genes ordered by temporal specificity (y-axis). Top shows sciLaMA-based pseudotime, bottom shows scVI results.

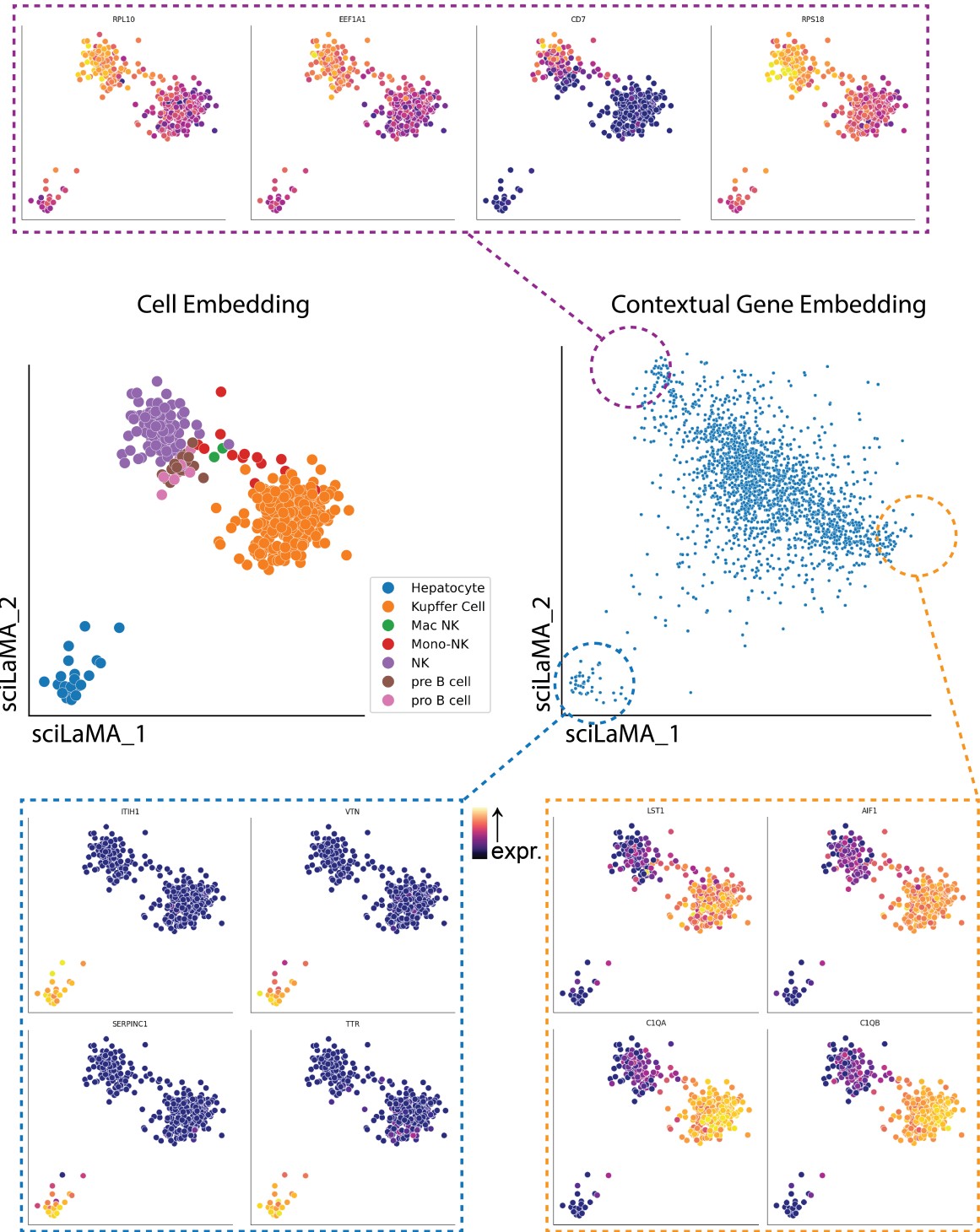

*Figure S4.* This visualization presents results from sciLaMA (with latent dimensionality $K$=2) trained on an exemplar fetal liver dataset (Choi et al., 2023). The cell embeddings (middle-left; dots represent individual cells) are colored according to annotated cell types, while the contextual gene embeddings (middle-right; dots represent individual genes) show corresponding embedding dimensions. The top and bottom panels illustrate cell embeddings colored by expression levels of genes sampled from distinct regions of the contextual gene embedding space. We quantitatively assessed sciLaMA's interpretability by computing feature attribution scores using Integrated Gradients applied to the pretrained cell VAE encoder. This resulted in a gene-by-latent-node attribution matrix, which we summarized into a single gene vector by aggregating absolute attribution scores. To facilitate comparison, we projected sciLaMA's contextual gene embeddings into a comparable vector form using vector normalization. The strong correlations (Pearson $r = 0.43$, Spearman $\rho = 0.46$) indicate that sciLaMA effectively captures key gene features in alignment with traditional stepwise attribution methods, yet does so within a more efficient, unified framework.

