# OpenReview forum: "sciLaMA: A Single-Cell Representation Learning Framework to Leverage Prior Knowledge from Large Language Models"
_ICML.cc/2025/Conference — ICML 2025 poster_

### Official Review · Reviewer_H2Qr · 2025-03-12

**Overall Recommendation:** 3

**Summary:**

The paper introduces a VAE approach to integrate external gene knowledge from pretrained large language models (LLMs) with single‐cell RNA sequencing (scRNA‐seq) data. By adapting static gene embeddings (derived from multiple modalities such as textual descriptions, protein sequences, and single‐cell models) into a context‐aware paired-VAE architecture, sciLaMA generates latent representations for both cells and genes. This combined embedding is then used for single-cell downstream tasks including cell clustering, batch effect correction, gene expression imputation, marker gene identification, and trajectory analysis. Benchmarking on several datasets demonstrates that embeddings from sciLaMA outperforms other approaches in terms of predictive accuracy.

**Claims And Evidence:**

Paper provides evidence for improvements in cell-level clustering, gene imputation, and computational efficiency compared to existing approaches. However, the claims regarding enhanced biological interpretability and broad generalizability are less convincingly supported, as they rely mainly on qualitative visualizations and indirect metrics without robust independent validation.

**Essential References Not Discussed:**

not the one I am aware of

**Experimental Designs Or Analyses:**

Yes I reviewed the experimental designs. Overall, these include multiple benchmark datasets and relevant quantitative metrics, but potential issues include the reliance on indirect measures of interpretability, sensitivity to hyperparameter tuning in the multi-stage training process, and a relatively narrow dataset scope that may limit generalizability.

**Methods And Evaluation Criteria:**

The proposed method and evaluation criteria are mostly suitable. However, the evaluation of biological interpretability remains somewhat indirect and might benefit from further targeted validation.

**Other Comments Or Suggestions:**

Typos or other errors:
In several places, terms like “large Language Models” are used inconsistently (“large language models” or “LLMs” or “large LaM”, “LaM”); it is preferable to use them consistently.
Line 156 page 3 the gene-level quantities are indexed with “j” and sometimes with “i” (e.g. “μ_gene_i”). Likewise in step 2. This appears to be a notational inconsistency that could confuse readers regarding whether the reconstruction error is computed per cell or per gene.

While not an error per se, the paper would benefit from a clearer, more unified notation that explicitly distinguishes between cell-level and gene-level variables throughout the loss functions.

For consistency, “cell by gene ” → “cell‐by‐gene “

Line 99 on the right side, it is unclear what “adapts and contextualizes” precisely means in this setting.

Line 207 page 4, “correspond meaningfully” is not quantified. It is unclear how one would assess whether a specific dimension of the cell latent space aligns with that of the gene latent space, or what “meaningfully” means in a rigorous sense.

“aligning each dimension” is not elaborated upon. Does this imply a one-to-one correspondence between latent dimensions, or does it involve a more complex mapping? More detail would help clarify the author's intention.

Line 259 and 260 make it clear “self-informed” → s.i.

sciLaMA (s.i.) is not immediately obvious how this variant is constructed or what “self-informed” entails compared to the other variants that integrate external knowledge.

**Other Strengths And Weaknesses:**

### Strengths:

By incorporating precomputed static gene embeddings from diverse modalities (text, protein sequence, and single-cell foundations), the framework enriches gene representation beyond what traditional VAEs (such as scVI or siVAE) offer.

The paired encoder-decoder architecture for both cells and genes allows sciLaMA to jointly learn context-aware representations.

Experiments on multiple tasks with extensive evaluations.

Pseudotime analysis demonstrates its utility in uncovering biologically relevant gene modules and temporal dynamics.

### Weaknesses

The motivation to combine features from a VAE with those from large language models (LLMs) seems somewhat arbitrary. One could argue that simpler methods, such as matrix factorization or even using siVAE-derived representations combined with LLM embeddings, might suffice. It is not clear what is an added value of using a VAE based probabilistic latent space.

The process of adapting static, fixed-length gene embeddings into a context-aware space through a paired-VAE is complex. The paper would benefit from a more detailed discussion on the robustness of the embedding alignment process, especially in scenarios where gene annotations or external descriptions are noisy or incomplete.

The framework depends heavily on static gene embeddings produced by LLMs. These embeddings capture information from texts, protein sequences, or prior single-cell studies, but their relevance to the intricacies of gene regulation and cellular context may be limited. In other words, the external knowledge may not always align perfectly with the biological nuances present in scRNA-seq data.

Furthermore, the quality and bias of these pretrained embeddings can propagate into the sciLaMA model, potentially skewing downstream interpretations if the external source is not well-calibrated for the target biological context.


The evaluation focus remains on specific tissues and conditions. It is unclear how well sciLaMA generalizes across other biological contexts or more heterogeneous datasets. A broader set of experiments could reinforce claims of universality.

Does not thoroughly explore the sensitivity of sciLaMA to key hyperparameters (e.g., latent space dimensionality, KL-divergence weights, alignment loss weight γ).

 The framework is presented as yielding “biologically interpretable” gene modules and cell representations. However, the review of marker gene identification and trajectory analysis relies primarily on visualizations and cluster-level metrics. Perhaps, a quantitative measure?

**Questions For Authors:**

Already covered above

**Relation To Broader Scientific Literature:**

sciLaMA builds upon the foundational work of VAEs in single-cell analysis. It addresses a key limitation of these models (the inability to integrate external gene knowledge) by incorporating pretrained gene embeddings from LLMs.

**Theoretical Claims:**

No theory in the paper.

---

> ### Author Rebuttal · Authors · 2025-04-01
>
> We address each of the concerns below with response figures and tables: https://github.com/anonymous-ICML2025/rebuttal_April1st/tree/main/H2Qr:
>
> **Interpretability**:
> The reviewer is correct that interpretability primarily relies on implicit gene module detection and clustering in the current manuscript. The alignment regularization term in our objective function is designed to align each dimension of cell and gene embeddings to ensure meaningful correspondence between them. In doing so, we are ‘encouraging’ genes with high weight in a particular dimension of the gene embeddings to explain variation in cell embeddings in the corresponding dimension (Response Fig 1).
>
> **Dataset scope**:
> We included a diverse set of datasets in this study, including human pancreatic, mouse spatial, human PBMC, and mouse neuronal datasets. We understand that expanding the dataset scope could further strengthen our findings, so in the revision we further incorporated additional datasets [See 5].
>
> **Sensitivity to hyperparameters**:
> In the revision, we tested a range of hyperparameter values [See 6].
>
> (1) “Matrix factorization method might suffice.”
>
> We did an experiment using a factorization-based approach in response to another reviewer, and showed sciLaMA is better performed (Reviewer 7H8M, Q2). Regarding using siVAE representations with LLM embeddings, this is conceptually similar to what sciLaMA is doing, as we clarified with one of the previous reviewers. sciLaMA can be considered as similar to siVAE but starting with LLM embeddings instead of learning them only from the training scRNAseq data.
>
> (2) “Detailed discussion on the robustness of the embedding alignment process when the external gene embeddings are noisy or incomplete.”
>
> This is a great point, which is why we included multiple types of external gene embeddings across different modalities. For instance, text-based gene embeddings are derived from ChatGPT responses or NCBI GeneCards summaries, protein sequence embeddings are derived from protein sequences through ESM and ProtTrans, and for single-cell foundation models, we utilize both scGPT and CellPLM to get static gene embeddings. To assess robustness, we evaluated sciLaMA using embeddings from all 3 modalities. The relatively small performance variance across these variants, as seen in our results, suggests sciLaMA is robust to noise in the input LLM embeddings. We also did experiments with additional hyperparameter choices for the alignment process to further validate this robustness [See 6].
>
> (3) “The external gene knowledge may not always align perfectly with the biological nuances present in scRNAseq data.”
>
> We agree, it is a limitation of external LLMs, but we consider this a strength of sciLaMA. Different biological contexts may require different knowledge sources, and sciLaMA allows users to explore multiple modalities and determine which functional context is most relevant to their dataset. We view our approach as a hypothesis testing strategy for checking which context is most important.
>
> (4) “Quality and bias of these pretrained embeddings can propagate into the sciLaMA model.”
>
> As mentioned above, sciLaMA provides flexibility in choosing external gene embeddings from different modal-specific LLMs, which enables users to select embeddings most relevant to their dataset, which overcomes potential biases introduced by any single knowledge source.
>
> (5) “Evaluation focus remains on specific tissues and conditions, unclear to show how well it can be generalized to other contexts.”
>
> In this manuscript, we have included data from diverse sources as described in [Dataset scope], and now further added comprehensive evaluations on two more datasets across diverse methods. We demonstrated sciLaMA robustly performs well in these datasets too (Response Fig 2-3, Repones Tables 1-2)
>
> (6) “Sensitivity of sciLaMA to hyperparameters such as latent space dim, KLD weight, and gamma.”
>
> For KLD, we follow the standard VAE framework and apply a beta value of 1, with a beta warm-up mechanism during training.
> For gamma, we set it to a relatively small value to prevent the alignment regularization term from dominating the loss function. Below, we present results using different gamma values (Response Fig 2, Response Table 3).
> For latent dim, we also evaluate performance across different values of $K$, as shown in the Response Fig 3, Response Table 4.
>
> (7) “Gene identification and trajectory analysis rely primarily on visualizations and cluster-level metrics. Provide a quantitative measure.”
>
> We have included quantitative validation metrics for gene module identification (Table S7). For gene trajectory analysis, we applied the Curvature-Based Smoothness Score (Response Table 5), showing after alignment, gene trajectories are more consistent than before.
>
> We also appreciate the reviewer’s careful attention to writing consistency and notation. We will thoroughly revise the manuscript to improve clarity and ensure consistency.

---

> > ### Comment · Reviewer_H2Qr · 2025-04-05
> >
> > I appreciate authors for preparing thorough response to my comments including additional experiments. However, I still have concerns and maintain my original assessment.
> >
> > Interpretability via alignment alone remains indirect and somewhat opaque. Perhaps a quantitative measure of biological interpretability or functional enrichment directly attributable to this alignment might be useful.
> >
> > The authors argue that the flexibility of choosing different modalities mitigates embedding bias. While flexibility is good, it does not fully solve the bias issue. There remains a risk that biases intrinsic to popular gene embedding methods (such as certain biological contexts being overrepresented in training data of foundation models) propagate into the final embeddings.
> >
> > While demonstrating superiority over a single factorization method is beneficial, comparisons with additional baseline methods (e.g., sparse VAE, ICA, PCA, other latent-space embedding approaches widely used in single-cell analyses) remain limited.

---

> > > ### Author Response · Authors · 2025-04-08
> > >
> > > We appreciate the reviewer's feedback. Below we address the remaining concerns, and we are happy to provide additional clarifications if needed.
> > >
> > > (1) Based on the fetal liver dataset results from the last round, to quantitatively evaluate interpretability, we computed feature attribution scores using Integrated Gradients on the cell VAE encoder (gene-by-latent-node matrix, then aggregated by absolute values) and compared these with sciLaMA's contextual gene embeddings (gene-by-latent-node matrix, then projected to a comparable shape via vector norms).
> > > The strong correlations (Pearson=0.43, Spearman=0.46) demonstrate that sciLaMA captures important gene features consistent with traditional stepwise attribution methods, while being more efficient with a single streamlined framework.
> > >
> > > (2) Regarding embedding bias, we acknowledge this is a fundamental challenge for all methods using pretrained models. While sciLaMA's flexible modality selection helps mitigate some biases, we agree this remains a limitation generally applied to frameworks that leverage prior knowledge.
> > >
> > > (3) We have added comparisons to PCA, ICA, and sparse VAEs (sparsity weights 0.1-0.5) as suggested by the reviewer in addition what we had in our original manuscript. sciLaMA maintains superior performance, which demonstrates advantages in modeling nonlinear gene relationships and structured sparsity. The results shown below will be included in the revision.
> > >
> > > |Methods  |PCC (↑)  |SCC (↑)	|JSD (↓)	|RMSE (↓)|
> > > |---|---|---|---|---|
> > > |sciLaMA-random (i)	          |0.051±0.027	|0.049±0.031	|0.289±0.009	|1.374±0.020|
> > > |sciLaMA-shuffled (ii)	      |0.056±0.036	|0.043±0.037	|0.288±0.009	|1.366±0.027|
> > > |sciLaMA-LLM (avg.)	          |0.222±0.027	|0.217±0.028	|0.283±0.008	|1.242±0.022|
> > > |MOFA+	                      |0.173±0.025	|0.192±0.023	|0.466±0.036	|1.282±0.019|
> > > |CellPLM-imputation	          |0.176±0.020	|0.104±0.029	|0.287±0.008	|1.281±0.016|
> > > |PCA	                        |0.177±0.026	|0.199±0.027	|0.297±0.008	|1.278±0.020|
> > > |ICA	                        |0.177±0.026	|0.199±0.027	|0.297±0.008	|1.278±0.020|
> > > |sparse VAE (sparsity = 0.1)	|0.201±0.025	|0.207±0.026	|0.299±0.011	|1.260±0.020|
> > > |sparse VAE (sparsity = 0.2)	|0.201±0.025	|0.207±0.026	|0.299±0.011	|1.260±0.020|
> > > |sparse VAE (sparsity = 0.3)	|0.201±0.025	|0.207±0.026	|0.299±0.011	|1.260±0.020|
> > > |sparse VAE (sparsity = 0.4)	|0.199±0.025	|0.204±0.027	|0.299±0.011	|1.261±0.020|
> > > |sparse VAE (sparsity = 0.5)	|0.200±0.025	|0.204±0.027	|0.300±0.011	|1.261±0.020|

---

### Official Review · Reviewer_mrfK · 2025-03-13

**Overall Recommendation:** 1

**Summary:**

The authors propose a new method that combines gene embeddings from LLMs with scRNA-seq data to generate context-aware representations for cells and genes. This approach outperforms existing methods in batch effect correction, cell clustering, and gene module discovery, while remaining computationally efficient. By incorporating biological knowledge, sciLaMA enhances single-cell analysis and improves interpretability in gene expression studies.

**Claims And Evidence:**

The authors do not provide enough evidence to illustrate that the text information to the cell representations. They only evaluated that text information can enhance the gene expressions in the clustering task, and with text information, it is worse than without in ASW metrics.

**Essential References Not Discussed:**

single cell with prior knowledge have been considered in [1]

[1] Tang W, Liu R, Wen H, et al. A general single-cell analysis framework via conditional diffusion generative models[J]. bioRxiv, 2023: 2023.10. 13.562243.

**Experimental Designs Or Analyses:**

I’ve checked the experiments. The experiments are not well designed. For example, the efficiency of cell representation should be evaluated by zero-shot setting in the cell classification, as the clustering task is weak in showing the results. Besides, it is wired that other baseline models are setting zero-shot, but the proposed method is trained. The following experiments are still weak. For example, in the spatial transcriptomic data comparison, they only include the non-pretrained models. However, at least a pretrained model, CellPLM, also includes the spatial transcriptomic data in the pertaining data, which needs to be considered. Similar to Table 4, it is too weak to only compare with scVI.

**Methods And Evaluation Criteria:**

The paper does not mention how to embed the molecular functions, but they show it in the framework figure. The experiments do not show the efficiency of the molecular functions.

**Other Comments Or Suggestions:**

no

**Other Strengths And Weaknesses:**

no

**Questions For Authors:**

no

**Relation To Broader Scientific Literature:**

The contribution is limited. The reconstruction framework, like scVI and CellPLM, is widely used in the single cell field. The authors claim that they can use the prior knowledge to enhance the performance, but the experiments are not strong enough to support their claims.

**Theoretical Claims:**

The paper does not have the theoretical claims.

---

> ### Author Rebuttal · Authors · 2025-04-01
>
> The reviewer mentioned that we did not provide enough evidence to illustrate how text information contributes to cell representations. In the manuscript we showed text-based gene information helped with removing batch effects (while preserving biologically meaningful information through sciLaMA framework). We are uncertain about the criticism “They only evaluated that text information can enhance the gene expressions in the clustering task”, could the reviewer kindly elaborate further?
>
> For the criticism “with text information, it is worse than without in ASW metrics”, in the single cell field, it’s widely acknowledged that no single metric holistically captures good performance and multiple metrics are often used for evaluation. As other reviewers have motioned, our work is evaluated on a wide range of downstream tasks and contains experiments on multiple tasks with extensive evaluations. We beat other methods in cell type clustering accuracy tasks (ARI and NMI) and integration tasks (batchASW, and iLISI), only performing worse than fine-tuned scGPT in the ASW metric (cell type separation). The fact that fine-tuned scGPT does a great job on cell type separation alone does not imply good overall performance because a method can high ASW when it separates both cell types and batches, which is undesirable (batches should be well mixed). That is why we measure both batchASW and iLISI to evaluate the integration performance more holistically and keep a good balance between removing batch effects (batches should not be separated) while preserving biologically meaningful information (cell types should be well separated).
>
> To the comment “the paper does not mention how to embed the molecular functions, but they show it in the framework figure”. We clarified in the manuscript that such information is derived from gene text descriptions, either sourced from ChatGPT-like LLMs responses, or NCBI GeneCard summaries (similar to GenePT and scELMo). If the reviewer is specifically referring to line 350 (right column, page 7), there obtained molecular function data directly from the Gene Ontology (GO) database.
>
> The review also criticized the experimental design by stating “the efficiency of cell representation should be evaluated by zero-shot setting in the cell classification”. We did not perform cell classification using zero-shot learning because the approach is inherently biased, and instead we followed well-established benchmarking frameworks including scIB (https://doi.org/10.1038/s41592-021-01336-8), as well as what have been mentioned in https://doi.org/10.1186/s13059-019-1850-9 by Tran et al. Both studies indicate strong rationales for using our selected metrics. We are not aware of any publication that explicitly argues for zero-shot cell classification, but would be happy to review any such papers and learn further insights if the reviewer has some in mind. We also looked at the scDiff paper by Tang et al., which the reviewer suggested us to cite. This paper covered “one-shot cell type annotation”, where the prior knowledge of cell type was utilized. This makes us unsure how “zero-shot setting in the cell classification” fits to our study, as our model does not incorporate cell type prior knowledge, and “zero-shot classification” is not one of our goals in this work.
>
> Another critique was that “baseline models are setting to zero-shot”, which is not true, as we included fine-tuned scGPT in our main figure and table. The reason why we did not fine-tune CellPLM was because running their code yields the error 'Currently CellPLM only supports zero shot embedding instead of fine-tuning’.
>
> For the spatial experiments, the reviewer stated that “they only include the non-pretrained models”. But every benchmarked model in this part was trained on the same dataset. To address the reviewer’s concern, we now compare against the pretrained foundation model CellPLM spexifically, and have added the results (**Response Table**: https://github.com/anonymous-ICML2025/rebuttal_April1st/tree/main/mrfK).
>
> For the reason why we compared sciLaMA and scVI in Table 4, please refer to (Reviewer 7H8M, Q3), and we provided more methods there.
>
> The reviewer stated, “The authors claim that they can use prior knowledge to enhance performance, but the experiments are not strong enough to support their claims.” Could the reviewer kindly clarify what would constitute strong enough evidence?  In one of our head-to-head comparisons, we evaluate models with and without prior knowledge using the same framework. Our results demonstrate that incorporating prior knowledge improves both performance (Tables 1, S6, S4, S7, Figure 2) and computational efficiency (Table 2).
>
> Finally, the reviewer also mentioned that the scDiff paper is one essential reference we did not discuss, we would like to cite this paper in our revision.

---

### Official Review · Reviewer_zJyw · 2025-03-15

**Overall Recommendation:** 4

**Summary:**

This paper proposes a new method for single cell embedding that leverages external textual information (or internal knowledge representation from large language models). The method works by considering a learnable matrix decomposition of the single cell data, where cell embeddings are produced from the transcriptomics data and the gene embeddings are produced from textual data. The authors then evaluate their methods on a wide range of downstream tasks, such as cell type clustering, gene expression reconstruction, marker gene identification, and trajectory analysis.

**Claims And Evidence:**

The results shown in this paper suggest sciLaMA outperforms previous state of the art models on a large scope of tasks.

**Essential References Not Discussed:**

NA.

**Experimental Designs Or Analyses:**

The cell-level analyses are established benchmarks. The gene reconstruction task is also standard in the spatial transcriptomics literature.

**Methods And Evaluation Criteria:**

Evaluation criteria and experimental methods make sense.

**Other Comments Or Suggestions:**

See above.

**Other Strengths And Weaknesses:**

Strengths:

The paper addresses an interesting challenge of leveraging large language models in single cell representation learning. The authors also produced an in-depth analysis of their method.

Weaknesses:

- Despite noting the inspiration from siVAE, the authors do not compare against that method. Could the authors explain why ?
- The sciLaMA-si is not clearly explained. What input is used in this case ? In my opinion, the most meaningful baseline would be to use a learnable gene embedding using the same architecture. That is, making $\mathbf{g}_j$ learnable, everything else being equal. This would represent the most direct assessment of the impact of external knowledge base.

**Questions For Authors:**

1. Despite noting the inspiration from siVAE, the authors do not compare against that method. Could the authors explain why ?

2. The sciLaMA-si is not clearly explained. What input is used in this case ?  Could the authors clarify this here and in the text ?

3. In my opinion, the most meaningful baseline would be to use a learnable gene embedding using the same architecture. That is, making $\mathbf{g}_j$ learnable, everything else being equal. This would represent the most direct assessment of the impact of external knowledge base. Could the authors argue against this, or compare against it if indeed relevant ?

**Relation To Broader Scientific Literature:**

This paper extends previous works in single cell embedding such as scVI and foundation models for single cell data such as scGPT.

**Theoretical Claims:**

There were no theoretical claims in this paper.

---

> ### Author Rebuttal · Authors · 2025-04-01
>
> Below are our responses to the questions:
>
> (1) “Why do you not use siVAE for comparison?”
>
> We now clarify this more explicitly in the manuscript. siVAE is functionally equivalent to sciLaMA-self-informed (sciLaMA-s.i.), which instead of using LLM-based external gene embeddings as the input to the gene encoder, sciLaMA-s.i instead uses the transposed gene expression matrix as input to the gene encoder. The two models share similar architectures, with some implementation differences: siVAE is built on TensorFlow, while sciLaMA is implemented in PyTorch. In addition to this, siVAE’s cell decoder produces probabilistic outputs (both mean and variance for reconstructing scaled gene expression), whereas sciLaMA’s cell decoder outputs only the mean with the assumption of unit variance after scaling. We introduced sciLaMA-s.i. as a non-LLM-based version specifically to reproduce siVAE, and we have clarified this more clearly in the revised manuscript.
>
> (2) “Explain sciLaMA-s.i. more clearly.”
>
> As addressed in response to question (1), sciLaMA-self-informed (sciLaMA-s.i.) uses the transposed gene expression matrix as input to the gene encoder instead of LLM-based external gene embeddings. Based on the reviewer’s suggestion, we have revised the manuscript to provide a clearer explanation of this.
>
> (3) “Try using a learnable gene embedding with the same architecture.”
>
> If we understand correctly, the reviewer is suggesting replacing the LLM-based external gene embeddings with randomly initialized values and allowing the model to learn contextual gene embeddings from scratch. If so, this question aligns with a similar request from another reviewer (Reviewer 7H8M, Q1), and we have included the corresponding results in our response there (**Response Table**: https://github.com/anonymous-ICML2025/rebuttal_April1st/tree/main/zJyw). If this is not what the reviewer intended, we would appreciate further explanation so that we can properly address the question and plan additional experiments if needed.
>
> **Response Table** to answer `Weaknesses` and `Questions For Authors` Q3
> |Methods| PCC (↑) | SCC (↑) | JSD (↓) | RMSE (↓) |
> | ------------- | ------------- | ------------- | ------------- | ------------- |
> | sciLaMA-random initialization| 0.051±0.027 | 0.049±0.031 | 0.289±0.009 | 1.374±0.020 |
> | sciLaMA (avg.)	|0.222±0.027|	0.217±0.028	|0.283±0.008|	1.242±0.022|

---

> > ### Comment · Reviewer_zJyw · 2025-04-04
> >
> > I thank the reviewer for addressing all my concerns and providing the extra experiment I had requested. I confirm I support accepting the paper.

---

> > > ### Author Response · Authors · 2025-04-08
> > >
> > > We sincerely appreciate the feedback, which helped improve our work. We are also grateful for the reviewer’s time and support.

---

### Official Review · Reviewer_7H8M · 2025-03-17

**Overall Recommendation:** 4

**Summary:**

The paper introduces sciLaMA, a novel framework for single-cell RNA sequencing analysis that integrates gene embeddings from large language models (LLMs) with scRNA-seq data using a Variational Autoencoder based architecture. This approach allows for context-aware representations of both cells and genes, enhancing tasks such as batch effect correction, cell clustering, imputation and gene module identification. The framework demonstrates superior performance in various single-cell analysis tasks, offering a flexible and interpretable solution for different single-cell tasks.

**Claims And Evidence:**

The claims the paper is making, which are:

"(1) We introduce a novel framework that incorporates diverse, external gene knowledge from pretrained LaMs with scRNA-seq data, facilitating context-aware cell and gene representation learning"
"(2) We demonstrate that our approach reduces computational requirements while improving performance compared to existing state-of the-art methods across various single-cell tasks"

are both underlined by the provided experiments.

**Essential References Not Discussed:**

%

**Experimental Designs Or Analyses:**

- Figure 3c: Could you provide a zoom-in side by side comparison of the imputation task, to present the differences a bit more nuanced and see what/how was imputed?
- Figure 5b: did you provide the time as an input to any of the models? (IP or EN, i.e. 0 or 1)

**Methods And Evaluation Criteria:**

- the proposed objective function seems solid. I believe the inference task was split into three training steps to improve robustness.
Three further comments regarding the experiments:
1) Robust cell representation learning and integration: It would be interesting to see how sciLaMA performs if the gene embeddings are random vectors (instead of none or LLM-based).
2) Imputation task: it would be interesting to see a simple kNN and/or linear approach as a benchmark as well. E.g., a simple linear factor analysis based approach such as MOFA [https://www.embopress.org/doi/full/10.15252/msb.20178124] also in the context of the gene set enrichment analysis, the imputation and the embeddings.
3) There are much more advanced tools than scVI to estimate pseudotime. Why did you go with scVI here?

**Other Comments Or Suggestions:**

- "discvovery" -> "discovery"
- "that across" -> "across"

**Other Strengths And Weaknesses:**

- I guess `h_j` is also of dimension l? Same for `W_{cell}`, whis is of dimension `l \times M`? Clarifying this early in the manuscript helps to understand the paper quicker.

**Questions For Authors:**

Please see questions above

**Relation To Broader Scientific Literature:**

%

**Theoretical Claims:**

There are no proofs, but also no necessity for any.

---

> ### Author Rebuttal · Authors · 2025-04-01
>
> Below, we address the three major points raised:
>
> (1) “How does sciLaMA perform if the gene embeddings are random vectors?”
>
> Using random gene vectors will prevent the model from generalizing to unseen genes. To further clarify this point, we conducted additional experiments: (i) using random gene embeddings as suggested by the reviewer, and (ii) randomly shuffling individual features within LLM-derived external gene embeddings for each gene. The results (Response Table 1) demonstrate that random external gene embeddings significantly degrade performance, which reflects the utility of LLM-derived external gene embeddings.
>
> (2) “Compare against a simple kNN or linear factorization method like MOFA for the imputation task.”
>
> We have included the imputation results using MOFA+ in the same table (**Response Table 1**). The results are not competitive with sciLaMA models.
>
> (3) “What is the reason why you used scVI in the pseudotime estimation?”
>
> Pseudotime inference (Section 4.4, Figure 5) was performed using the Palantir framework, but requires cell embeddings to be pre-computed as input. The goal of pseudotime inference in our manuscript was not to show that “sciLaMA is the best performing pseudotime inference method”, but to show “incorporating external gene information via LLMs into existing approaches improves their pseudotime inference”. We chose scVI because of its VAE-based inference, which matches sciLaMA’s VAE based inference, with the key distinction being that sciLaMA integrates gene information from LLMs. To further address this concern, we now included additional methods here to learn cell embeddings (**Response Table 2**).
>
> Regarding other figure-related comments:
>
> Figure 3c: We have provided a zoomed-in version at the following anonymous link (**Response Figure**:
> https://github.com/anonymous-ICML2025/rebuttal_April1st/tree/main/7H8M) and will include this in the revision.
>
> Figure 5b: No developmental time or cell state information was provided to any model. This ensures that differences in developmental states are captured only from gene expression and gene embeddings, which maintains a fair comparison.
>
> For the rest of comments on notation and typos:
>
> (a) Yes, each gene’s $h_j$ has a dimensionality of $l$, which corresponds to $W^{cell}$ with dimensions $l \times M$. We will clarify this in the manuscript.
>
> (b) We have corrected the typos pointed out by the reviewer in the revision.
>
>
> **Response Table 1** to answer `Methods And Evaluation Criteria` Q1 & Q2
> |Methods| PCC (↑) | SCC (↑) | JSD (↓) | RMSE (↓) |
> | ------------- | ------------- | ------------- | ------------- | ------------- |
> | sciLaMA-random (i) | 0.051±0.027 | 0.049±0.031 | 0.289±0.009 | 1.374±0.020 |
> | sciLaMA-shuffled (ii) | 0.056±0.036 | 0.043±0.037 | 0.288±0.009 | 1.366±0.027 |
> |sciLaMA-LLM (avg.)	|0.222±0.027|	0.217±0.028	|0.283±0.008|	1.242±0.022
> | MOFA+ | 0.173±0.025 | 0.192±0.023 | 0.466±0.036	| 1.282±0.019 |
>
> **Response Table 2** to answer `Methods And Evaluation Criteria` Q3
> |Methods| ARI (↑) | NMI (↑) | ASW (↑) | cLISI (↑) |
> | ------------- | ------------- | ------------- | ------------- | ------------- |
> |scVI	| 0.284	|0.291	|0.501	|0.501 |
> |cellPLM |0.216	|0.281	|0.513	|0.684 |
> |GenePT-w	|0.000	|0.017	|0.409	|0.514 |
> |scGPT	|0.275	|0.310	|0.530	|0.716 |
> |scGPT-finetuned	|0.033	|0.061	|0.449	|0.535 |
> |sciLaMA-scGPT	|0.341	|0.374	|0.523	|0.738 |
> |sciLaMA-ESM	|0.299	|0.369	|0.530	|0.833 |
> |sciLaMA-ProtTrans	|0.355	|0.366	|0.529	|0.775|
> |sciLaMA-GenePT	|0.237	|0.321	|0.514	|0.776 |
> |sciLaMA-ChatGPT	|0.316	|0.351	|0.518	|0.738 |
> |sciLaMA-CellPLM	|0.343	|0.381	|0.524	|0.793|

---

> > ### Comment · Reviewer_7H8M · 2025-04-07
> >
> > (1) “How does sciLaMA perform if the gene embeddings are random vectors?” AND
> > (2) “Compare against a simple kNN or linear factorization method like MOFA for the imputation task.”
> >
> > Thanks for providing the updated results. I was expecting to see this, but was great to get this clarification.
> >
> > (3) “What is the reason why you used scVI in the pseudotime estimation?”
> >
> > Thanks for this clarificaiton as well.
> >
> > I believe this is a good contribution to the CompBio field by showing how domain knowledge (from LLMs) can be used. It would be great to see the next steps as well: showing that "causal" language knowledge (providing true gene-gene relationships) is more useful than "correlation-based" language knowledge. Happy to raise my evaluation to "Accept".

---

> > > ### Author Response · Authors · 2025-04-08
> > >
> > > We fully agree that disentangling causal vs. correlational gene relationships is an exciting direction. The supportive feedback is much appreciated.

---

### Decision · Program_Chairs · 2025-05-01

**Decision:**

Accept (poster)

**Comment:**

This paper presents sciLaMA, a novel framework for single-cell RNA sequencing analysis that integrates pretrained gene embeddings from large language models (LLMs) with a variational autoencoder-based architecture. The core idea, leveraging external textual and biological knowledge to improve context-aware representation of genes and cells, is timely. The method is evaluated across multiple downstream tasks, including cell clustering, gene imputation, and trajectory inference, and demonstrates competitive or superior performance compared to existing approaches.

While some concerns were raised regarding baseline comparisons, evaluation depth for interpretability, and broader generalizability, the authors have proactively addressed these in their rebuttal, providing additional experiments and clarifications. Remaining points for improvement, as emerged during discussion, are:
- a careful evaluation using classification tasks
- better description of the novelty, as previous work has already demonstrated the utility of injecting text information
- better articulate the rationale behind the chosen approach: why was this particular model adopted

Overall, this is a well-executed and promising contribution that will be of interest to the machine learning and computational biology communities. I recommend acceptance.